# Phylogenetically informed predictions outperform predictive equations in real and simulated data

Jacob D. Gardner [1,3], Joanna Baker [1,3], Chris Venditti [1] ✉ & Chris L. Organ [2] ✉

Inferring unknown trait values is ubiquitous across biological sciences—whether for reconstructing the past, imputing missing values for further analysis, or understanding evolution. Models explicitly incorporating shared ancestry amongst species with both known and unknown values (phylogenetically informed prediction) provide accurate reconstructions. However, 25 years after the introduction of such models, it remains common practice to simply use predictive equations derived from phylogenetic generalised least squares or ordinary least squares regression models to calculate unknown values. Here, we use a comprehensive set of simulations to demonstrate two- to three-fold improvement in the performance of phylogenetically informed predictions compared to both ordinary least squares and phylogenetic generalised least squares predictive equations. We found that phylogenetically informed prediction using the relationship between two weakly correlated (r = 0.25) traits was roughly equivalent to (or even better than) predictive equations for strongly correlated traits (r = 0.75). A critique and comparison of four published predictive analyses showcase real-world examples of phylogenetically informed prediction. We also highlight the importance of prediction intervals, which increase with increasing phylogenetic branch length. Finally, we offer guidelines to making phylogenetically informed predictions across diverse fields such as ecology, epidemiology, evolution, oncology, and palaeontology.

Prediction is the very heart of what defines science[1]. It flows directly from hypotheses and theories as the arbiter of evidence. Researchers predict trait values for various reasons. For example, we might want to impute missing values of a dataset intended for further analysis (e.g.[2–4]). We may also want to make general inferences to glean knowledge about the adaptation and evolution of trait variation amongst species. In evolutionary biology specifically, and historical sciences more generally, we are often interested in retrodictions—predictions about past events[5–7].

Whatever our motivations, phylogenetic comparative methods (PCMs) have revolutionised our understanding of evolutionary biology, offering profound insights into the patterns and processes shaping biodiversity[8–10]. Phylogenetic comparative methods also provide a principled way to predict unknown values[11]. Owing to common descent, data drawn from closely related organisms are more similar than data drawn from distant relatives[12]. Among PCMs, phylogenetically informed prediction using regression techniques has emerged as an essential tool[11–15] (see Box 1) to predict unknown values given both information on shared ancestry and an underlying evolutionary relationship between traits. For example, phylogenetically informed prediction has been used to predict the time spent feeding in extinct hominins using the relationship between feeding time and molar size

---

[1]School of Biological Sciences, University of Reading, Reading RG6 6AJ, UK. [2]Department of Earth Sciences, Montana State University, Bozeman, MT 59717, USA. [3]These authors contributed equally: Jacob D. Gardner, Joanna Baker. ✉e-mail: c.d.venditti@reading.ac.uk; organ@montana.edu

## BOX 1
# Phylogenetically informed prediction

In the context of ordinary least squares (OLS) regression, the relationship between the dependent variable (Y) and independent variables (X) is modelled with Eq. (1):

$$Y = \beta_0 + \beta_1 X_1 + \beta_2 X_2 + \ldots + \beta_n X_n + \varepsilon \tag{1}$$

where $\beta_0$ is the intercept and $\beta_1, \beta_2, \ldots, \beta_n$ are the coefficients for the independent variables. The error term $\varepsilon$, describes the residual variance. The coefficients are estimated by minimising the sum of the squared differences between the observed values and the values predicted by the model. Mathematically, this involves solving for $\beta$ in Eq. (2):

$$Y = X\beta + \varepsilon \tag{2}$$

where $Y$ is the vector of observed values, $X$ is the matrix of independent variables, and $\beta$ is the vector of coefficients. Once the model is fitted and the coefficients are estimated, predictions for the dependent variable can be made using Eq. (3):

$$\hat{Y} = \hat{\beta}_0 + \hat{\beta}_1 X_1 + \hat{\beta}_2 X_2 + \ldots + \hat{\beta}_n X_n \tag{3}$$

Phylogenetic generalised least squares regression (PGLS) extends the OLS framework by incorporating the phylogenetic variance-covariance matrix $V$ into the error term to account for the non-independence of observations. The coefficients in PGLS regression are estimated by again solving for $\beta$ in Eq. (2), but here $\varepsilon \sim N(0, V)$ and the GLS estimator $\hat{\beta} = (X^T V^{-1} X)^{-1} (X^T V^{-1} Y)$ accounts for the phylogenetic relationships[13].

Coefficients of both OLS and PGLS models are often used to generate *predictive equations*—from which unknown values of *y* are simply calculated given the value of the independent variable(s) ($y = \alpha + \beta x$).

However, *phylogenetically informed prediction* explicitly incorporates the phylogenetic position of the unknown species relative to those used to inform the regression model. In this scenario, phylogenetically informed predictions for a species *h* can be made using Eq. (4):

$$\hat{Y}_h = \hat{\beta}_0 + \hat{\beta}_1 X_1 + \hat{\beta}_2 X_2 + \ldots + \hat{\beta}_n X_n + \varepsilon_u. \tag{4}$$

These predictions use both the estimated coefficients and $\varepsilon_u = \left( V_{ih}^T V^{-1} \right)^{-1} (Y - \hat{Y})$, where $V_{ih}^T$ is a $n \times 1$ vector of phylogenetic covariances for all species *i* other than species *h*. Therefore, a *phylogenetically informed prediction* is computed by adjusting the prediction off the regression line by $\varepsilon_u$—a prediction residual. This method was first described by Garland & Ives[11] using independent contrasts on a tree re-rooted at the node of interest, with various implementations having been introduced since (see Box 2).

Below, we illustrate a hypothetical example using two correlated continuously varying traits (yellow and pink bars) related to the tips of a phylogenetic tree. The variance-covariance matrix $V$ includes the shared branch lengths between taxa (measured in proportion of total tree height). OLS (green line) and PGLS (orange line) regression methods can yield different slopes in the relationship. Green and orange points show estimates of the missing pink trait (?) when calculated from OLS and PGLS regression equations. However, by explicitly incorporating phylogenetic information, in this case the phylogenetically informed prediction (blue circle) pulls the estimate away from both calculations made by the predictive equations and closer to its sister taxa (grey points). Figure modified from Organ[7].

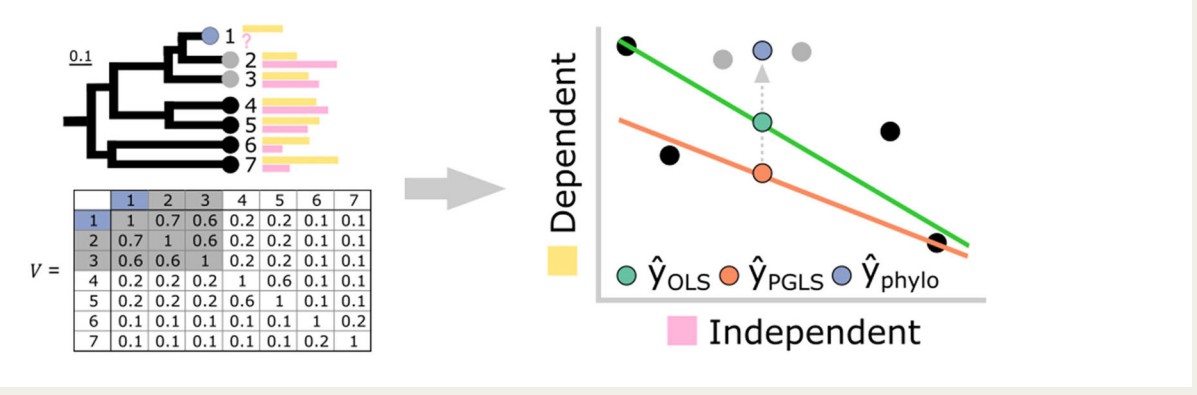

in living species (along with measurements from fossil individuals)[16]. Phylogenetically informed prediction explicitly addresses the non-independence of species data by calculating independent contrasts[12], using a phylogenetic variance-covariance matrix to weight data in phylogenetic generalised least squares[13,15] (PGLS), or by creating a random effect in a phylogenetic generalised linear mixed model[14] (PGLMM). Each of these approaches includes phylogeny as a fundamental component of the statistical model and thus yield equivalent results[11,14]. Organ et al.[6] further advanced phylogenetically informed prediction by developing a Bayesian application, which enabled the sampling of predictive distributions for further analysis and was implemented on extinct species for the first time. Importantly, the direct incorporation of phylogenetic relationships allows the option to predict unknown values from only a single trait, using the shared evolutionary history of the trait among known taxa. For example, it would be possible to predict the molar size of extinct species with no dental fossil record using extant variation in molar size along with a phylogenetic tree—this is, of course, only possible in the context of phylogenetically informed prediction. Methods for phylogenetically informed predictions have been used to great success: since their inception, we have seen genomic and cellular traits reconstructed for dinosaurs[6,17], a trait database built with phylogenetic imputation spanning tens of thousands of tetrapod species[2], and a map of the global geographical distribution of tree functional diversity[18].

However, 25 years on from the initial introduction of these methods[11] and despite the recognised pervasiveness of phylogenetic signal in continuous datasets[19,20], the use of predictive equations, which involve only the resulting coefficients of regression models (see Box 1), to calculate morphologies, behaviours, and ecologies of species dominates the literature. They persist to the extent that papers are devoted to providing predictive equations[21–27]—and even entire books have been dedicated to deriving predictive equations from allometric relationships[28,29]. Calculating unknown trait values from predictive equations (i.e., the regression coefficients) by themselves excludes information on the phylogenetic position of the predicted taxon. Using predictive equations in the absence of a predicted taxon's phylogenetic position still perseveres, despite the demonstration that phylogenetically informed predictions are likely to be much more accurate[11,30] and the knowledge that data produced by evolution without accounting for phylogenetic structure will suffer from pseudo-replication, misleading error rates, and spurious results[31,32]. In recent years, there has been an influx of papers using predictive equations from PGLS models in an attempt to account for shared ancestry (e.g.[21,33–35]). The parameters of a phylogenetic regression model are explicitly interpretable only in combination with the underlying phylogeny, and thus calculating unknown values using such predictive equations are also likely to be inaccurate and biased—and while it is probable that they are more so than non-phylogenetic regression equations (Box 1), this has never been explicitly demonstrated.

Here, we unequivocally demonstrate the superior performance of phylogenetically informed predictions compared to predictive equations derived from both OLS and PGLS regression models using an extensive set of simulations from ultrametric trees (where all species terminate at the same time) and non-ultrametric trees (where tips vary in time). We combine our simulations with application to four real-life datasets incorporating both living and fossil diversity: primate neonatal brain size, avian body mass, bush-cricket (katydid) calling frequency, and non-avian dinosaur neuron number. We provide a systematic critique by quantifying the prediction performance of all three approaches (phylogenetically informed prediction, OLS predictive equations, and PGLS predictive equations). We then provide a roadmap for using prediction within comparative studies in fields ranging from ecology and epidemiology to evolution and palaeontology.

## Results and discussion
### Simulations
**The performance of phylogenetically informed prediction on ultrametric trees.** Here, we assess the performance of phylogenetically informed predictions compared to OLS and PGLS-derived predictive equations under different evolutionary scenarios. Following the original formulation of phylogenetically informed prediction[11] (Box 1), we frame our analyses in the context of simple bivariate regression. However, it is important to note that these approaches can be generalised to incorporate any number of independent variables— and for phylogenetically informed prediction, this includes prediction from the phylogeny alone. We begin with a sample of 1000 ultrametric trees (Fig. 1a, Supplementary Fig. 1a, e), all with $n = 100$ taxa and with varying degrees of balance (Fig. 1a), reflecting real datasets (see case studies). Balance is the degree to which subsets of a tree are symmetrical in length or size[36]. For each tree, we simulate continuous bivariate data with three different correlation strengths ($r = 0.25$, $0.5$, and $0.75$) using a bivariate Brownian motion model[37], resulting in 3000 simulated datasets. We then predict the dependent trait value for 10 randomly selected taxa from each dataset using all three approaches and calculate prediction errors by subtracting predicted values from the original, simulated values. Additionally, we repeat this procedure for trees with 50, 250, and 500 taxa to account for and quantify the effect of varying tree size.

All three approaches across all trees and simulated correlation strengths have median prediction errors close to 0, suggesting low bias across methods (Fig. 1a–d; Supplementary Data 1). We calculate the variance ($\sigma^2$) of the prediction error distributions to summarise the overall performance of each method, where smaller $\sigma^2$ (narrower distributions) indicate that a method is consistently more accurate across the 1000 simulations and thus has greater overall performance. For ultrametric trees, phylogenetically informed predictions perform about 4–4.7× better than calculations derived from OLS and PGLS predictive equations (Fig. 1a–d)—that is, the $\sigma^2$ for phylogenetically informed prediction (e.g., $\sigma^2 = 0.007$ when $r = 0.25$) is about 4–4.7× smaller than that for predictions made from either OLS ($\sigma^2 = 0.03$ when $r = 0.25$) or PGLS ($\sigma^2 = 0.033$) equations (see Supplementary Data 1 for variances in error distributions). The improved performance of phylogenetically informed prediction is observed for each set of correlation coefficients, and performance naturally improves with more strongly correlated data. Phylogenetically informed predictions from only weakly correlated datasets ($r = 0.25$, $\sigma^2 = 0.007$) have about 2× greater performance even when compared to predictive equations from more strongly correlated datasets ($r = 0.75$, $\sigma^2 = 0.015$ and $0.014$ for PGLS and OLS predictive equations, respectively; Supplementary Data 1).

These distributions of errors encompass multiple predictions made from 1000 simulated phylogenies. The median prediction errors from such distributions are therefore not useful for comparing accuracy. To compare the prediction accuracy among the three approaches, we calculated the difference in the absolute prediction errors for each of the predictive equations and those for the phylogenetically informed predictions (i.e., error difference = absolute OLS or PGLS predictive equation error−absolute phylogenetically informed prediction error). If the error difference is positive, then the phylogenetically informed prediction has the more accurate prediction. The OLS and PGLS predictive equations are more accurate if the error difference is negative. In about 96.5–97.4% of the 1000 ultrametric trees, the phylogenetically informed predictions are closer to the actual values than the estimates from PGLS predictive equations (i.e., the median error difference of each tree is positive in 96.5–97.4% of trees). Phylogenetically informed predictions are more accurate than OLS predictive equations in 95.7–97.1% of trees. To test if these error differences statistically differ from 0, we performed intercept-only linear models (equivalent to one-sample t-tests) on the median error difference from each tree ($n = 1000$ median error differences each to

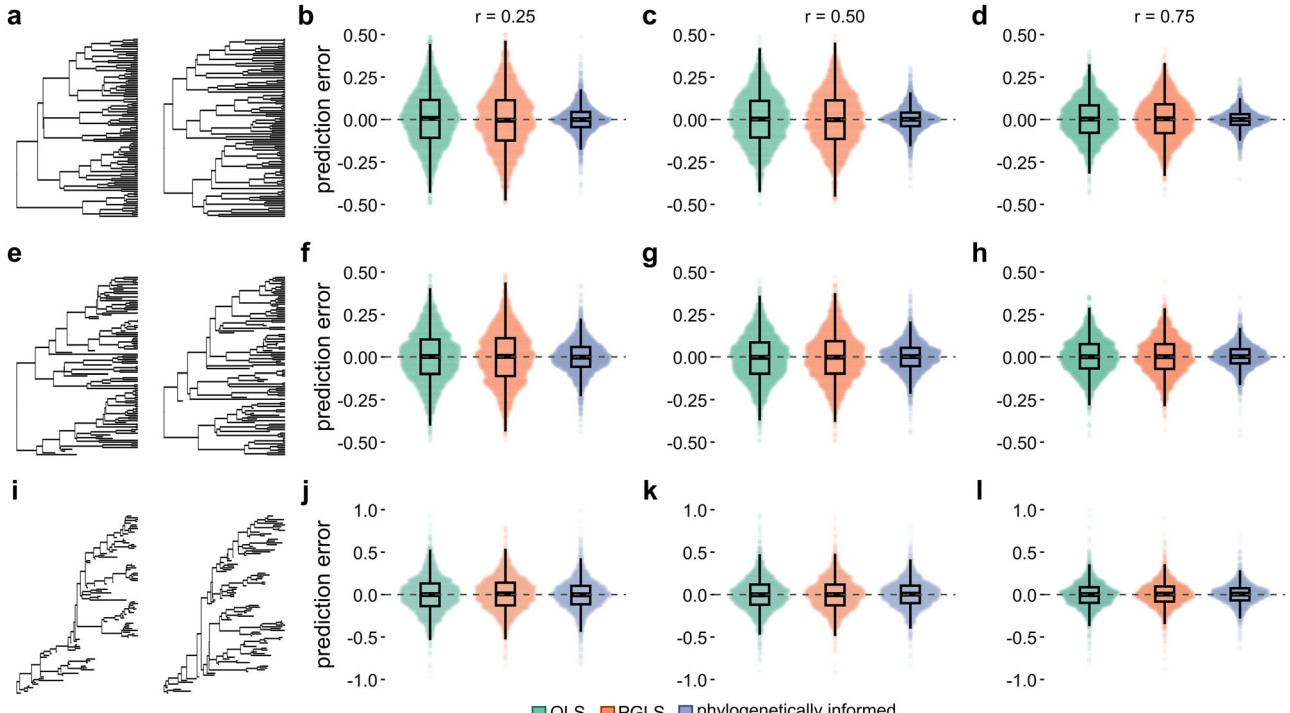

**Fig. 1 | Performance of phylogenetically informed prediction against predictive equations. a–d** Distributions of prediction errors (actual−predicted values) from a set of 1000 simulated ultrametric trees ($n = 100$ tips) under three correlation coefficients (r = 0.25, 0.5, and 0.75). **e–l** Distributions of prediction errors from 1000 non-ultrametric trees ($n = 100$ tips) with low (**e–h**) and high (**i–l**) extinction rates. Boxplots elements are as follows: centre line, median; box limits, first and third quartiles; whiskers, 1.5 × inter-quartile range; points, outliers. Colours represent the three prediction methods (OLS: ordinary least squares predictive equations, green; PGLS: phylogenetic generalised least squares predictive equations, orange; phylogenetically informed predictions, blue). Tighter distributions near zero indicate more accurate predictions overall than those more skewed away from zero (measured by the variance in prediction errors, $\sigma^2$). Phylogenetically informed predictions are more accurate than other methods. Medians and variances of distributions can be found in Supplementary Data 1.

compare the phylogenetically informed prediction errors with the PGLS and OLS predictive equation errors; see Methods for model details. Differences in the median prediction error between OLS and PGLS-derived predictive equations and phylogenetically informed predictions are positive on average across the 1000 ultrametric trees, which decrease with increasing correlation (average estimated error difference = 0.05–0.073, $p$-values < 0.0001; Supplementary Data 2). In other words, predictive equations have greater prediction errors and are less accurate than phylogenetically informed predictions.

The median absolute error in the phylogenetically informed predictions from each ultrametric tree is independent of tree stemminess defined as the accumulation of speciation events towards the tips ($p$-values > 0.05; Supplementary Data 3). The degree of tree balance explains less than 1% of the variation in the median absolute error of the phylogenetically informed predictions for two correlations (Colless's and Sackin's metrics: $p$-values < 0.05, adjusted $R^2$ < 0.006 and 0.003 when $r = 0.25$ and 0.75, respectively; Supplementary Data 3). Tree shape, including degree of balance and stemminess, does not affect the median absolute error in the PGLS predictive equations ($p$-value > 0.2). The predictive performance of all approaches improves as the number of taxa in the trees increases from 50 to 500, with phylogenetically informed predictions performing about 3.3–3.7× better than predictive equations in small trees ($n = 50$ taxa) to about 6–7× better in large trees ($n = 500$ taxa) (Supplementary Data 4). Phylogenetically informed predictions remain more accurate than predictive equations across different tree sizes, with the difference in median absolute error decreasing as the number of taxa increases (average estimated error difference = 0.053–0.087 when $n = 50$ taxa and 0.027–0.042 when $n = 500$ taxa, $p$-values < 0.0001; Supplementary Data 5); all methods improve accuracy with larger sample sizes.

We also simulated datasets for two end-member cases: a completely balanced tree (where each branch has equal numbers of terminal nodes, Supplementary Fig. 1a) and an imbalanced tree (where the number of species originating at each node changes through the tree, Supplementary Fig. 1e). For both trees, phylogenetically informed predictions perform about 3.5× and 1.2–1.5× better, respectively, than OLS or PGLS predictive equations across correlation strengths (Supplementary Fig. 1a–h; Supplementary Data 6). Phylogenetically informed prediction performs marginally worse in the imbalanced tree compared to the balanced one but has substantially greater performance than either predictive equation (Supplementary Fig. 1e–h). Both predictive equations also have greater prediction errors and are less accurate on average than phylogenetically informed predictions on completely balanced and imbalanced ultrametric trees (average estimated error difference = 0.008–0.047, $p$-values < 0.0001; Supplementary Data 7).

**Predicting extinct taxa.** To assess the performance of predicting trait values for extinct taxa, we simulate datasets from a distribution of 1000 non-ultrametric trees with 100 taxa under low and high extinction rates (Fig. 1e,i). We then predict all extinct taxa based only on the traits observed in extant taxa and compare the prediction errors among the three methods. These simulations are also repeated over tree sizes of 50, 250, and 500 taxa. Phylogenetically informed prediction performs about 2.4–2.7× better than predictive equations in trees with lower extinction rates (Fig. 1e–h) and performs better regardless of the simulated correlation coefficient (Supplementary Data 1). The phylogenetically informed predictions are closer to the actual values than the estimates from PGLS predictive equations in 91.5–92.7% of the 1000 non-ultrametric trees with low extinction. Phylogenetically informed predictions are also more accurate than OLS predictive equations in 91.9–93% of trees. Predictive equations

have higher absolute errors and are statistically less accurate on average than phylogenetically informed predictions (average estimated error difference = 0.026–0.044, $p$-values < 0.0001; Supplementary Data 2). The median absolute errors in the phylogenetically informed predictions are not associated with the degree of balance and stemminess ($p$-values > 0.05) except when data are strongly correlated ($r = 0.75$) where absolute error and tree balance are correlated; however, balance metrics explain only 0.77% of the variation in median absolute error (Colless's and Sackin's metrics: $p$-values = 0.0031, adjusted $R^2$ = 0.00774; Supplementary Data 3). The degree of tree balance explains ~ 0.5% to 1% of the variation in median absolute error of PGLS predictive equations (Colless's and Sackin's metrics: $p$-values < 0.05, adjusted $R^2$ = ~ 0.005 and 0.01 when $r = 0.25$ and 0.75, respectively; Supplementary Data 3). Stemminess explains only 0.41% of the variation in median absolute error of PGLS equations when the data are strongly correlated ($r = 0.75$; $p$-value = 0.023, adjusted $R^2$ = 0.0041; Supplementary Data 3). Phylogenetically informed predictions perform about 2× better than predictive equations on small trees ($n = 50$ taxa) to about 4× better on large trees ($n = 500$ taxa) (Supplementary Data 4) and are more accurate than predictive equations across different tree sizes (average estimated error difference = 0.029–0.044 when $n = 50$ taxa and 0.019–0.03 when $n = 500$ taxa, $p$-values < 0.0001; Supplementary Data 5).

The performance of all three methods reduces as simulated extinction level increases (Fig. 1i–l) owing to decreasing statistical power (i.e., fewer extant taxa to inform predictions). Many predictions in this scenario are phylogenetic extrapolations, a situation in which the target taxon falls outside the phylogenetic coverage of the extant taxa and therefore have limited shared evolutionary history with the extant taxa in the dataset. Regardless, a little phylogenetic information is better than none: phylogenetically informed prediction still performs 1.2–1.4× better overall than predictive equations (Fig. 1i–l; Supplementary Data 1). In 75.9–77.7% of 1000 non-ultrametric trees with high extinction, the phylogenetically informed predictions are closer to the actual values than the estimates from PGLS predictive equations. Phylogenetically informed predictions are also more accurate than OLS predictive equations in 75.5–78.8% of trees. These differences in absolute error are significantly greater than 0 on average (average estimated error difference = 0.014–0.026, $p$-values < 0.0001; Supplementary Data 2). In trees with high extinction, balance explains 0.3–3% of the variation in the median absolute error of phylogenetically informed predictions (Colless's metric: $p$-value < 0.05, adjusted $R^2$ = 0.003–0.03; Sackin's metric: $p$-value < 0.05, adjusted $R^2$ = 0.005–0.03 across datasets where $r = 0.25$, 0.5, and 0.75; Supplementary Data 3), while there is no association with stemminess ($p$-value > 0.05; Supplementary Data 3).

When we retain deeper evolutionary history by predicting 20% of all taxa (i.e., retaining dependent values for some extinct taxa), the performance of phylogenetically informed prediction in the high-extinction trees increases dramatically by about 5× overall ($\sigma^2$ in the distribution of phylogenetically informed prediction errors = 0.0078, 0.0063, and 0.0037 for $r = 0.25$, 0.5, and 0.75, respectively; Supplementary Data 8). Predictive equations perform relatively poorly ($\sigma^2$ for OLS = 0.034, 0.026, and 0.016, and PGLS = 0.062, 0.049, and 0.03; Supplementary Data 8). Predictive equations are also considerably less accurate in such scenarios, with calculations from PGLS equations being worse than OLS (average estimated error difference = 0.077–0.11 for PGLS equations and 0.044–0.067 for OLS, $p$-values < 0.0001; Supplementary Data 9). In an extreme-extinction scenario (pectinate or ladder-like tree, e.g., Supplementary Fig. 1i), performance is greatest when using phylogenetically informed prediction (Supplementary Data 6). However, there is a drastic reduction in the performance of predictions made from PGLS predictive equations compared to those derived from OLS equations. The PGLS equations are also much more inaccurate compared to the phylogenetically informed predictions

than OLS equations (average estimated error difference = 0.19–0.27 for PGLS equations and 0.075–0.11 for OLS, $p$-values < 0.0001; Supplementary Data 7). The PGLS regression line is estimated using shared ancestry. Using only the estimated PGLS regression parameters ignores the phylogenetic component of the model and therefore can severely misinform predictions (Box 1).

**Prediction uncertainty, prediction intervals, and branch lengths.** For all approaches, it is possible to assess the uncertainty around each individual prediction by calculating a *prediction interval*: a range of values within which an estimate is likely to fall given some criterion for confidence (e.g., 95%). A prediction interval is calculated using a combination of the underlying model variance and the expected variance in the prediction[11] (see Methods). In contrast to prediction intervals derived from OLS regression, the expected variance in the phylogenetically informed prediction accumulates with time, according to Brownian motion. As expected, and in line with previous simulation studies on ultrametric trees[30], phylogenetically informed prediction intervals increase with the target taxon's terminal branch length across all tree types (Fig. 2, Supplementary Fig. 2). The more time a taxon has had to diverge from its sister taxa, the less certain we are about its prediction. As stated by Garland and Ives[11] (pg. 350), *"... if the hypothetical species were attached infinitely close to the sister tip, then the prediction intervals would diminish to 0."* In line with this, we find that the range of prediction errors in the phylogenetically informed predictions also increases with terminal branch lengths across our simulations (Fig. 2, Supplementary Fig. 2). The more uncertain we are about a prediction, the less accurate we are likely to be. When divergences from sister taxa are shortest, phylogenetically informed prediction will tend to be more accurate (Box 1). Prediction intervals and prediction errors derived from OLS regressions do not vary with terminal branch lengths because phylogenetic information is not considered. Increasing prediction intervals with longer terminal branches is not a cause for concern—instead, it is necessary and important information to incorporate when making predictions. The level of certainty about a taxon's evolutionary history is entirely ignored by predictive equations.

The impact of branch lengths on the uncertainty of our estimates therefore highlights a potential problem. For example, the dominance of parsimony methods in palaeontological studies means that branch lengths are often not considered and autapomorphies are routinely ignored (these are used to determine terminal branch lengths). It is possible to compute a "generic" prediction interval from a PGLS regression without knowing a target taxon's phylogenetic position. In this case, the generic prediction interval effectively assumes the predicted taxon to be attached to the root of the tree with a terminal branch length equal to the average length of the branches leading from the root to each tip[11] (i.e., it is the phylogenetic average). However, we can improve on these generic prediction intervals and mitigate the lack of branch length information by setting all branch lengths to be equal across the entire tree (e.g., all branch lengths = 1; see Case study 3 below). Although an arbitrary assumption, using equal branch lengths retains the phylogenetic structure needed to account for shared ancestry in the predictions[11]. To assess the performance of this strategy, we took the same set of trees from the ultrametric (Fig. 1a) and non-ultrametric low-extinction simulations (Fig. 1e) and transformed all branches to equal a length of 1 (Supplementary Fig. 3). Even when assuming equal-length branches, phylogenetically informed predictions have greater overall model performance (Supplementary Data 10) and are more accurate (Supplementary Data 11) than both predictive equations.

In any context in which a prediction is to be made, it is therefore important that researchers present them alongside prediction intervals. Phylogenetically informed predictions provide us with estimates of values that are explicitly framed in an evolutionary context—the

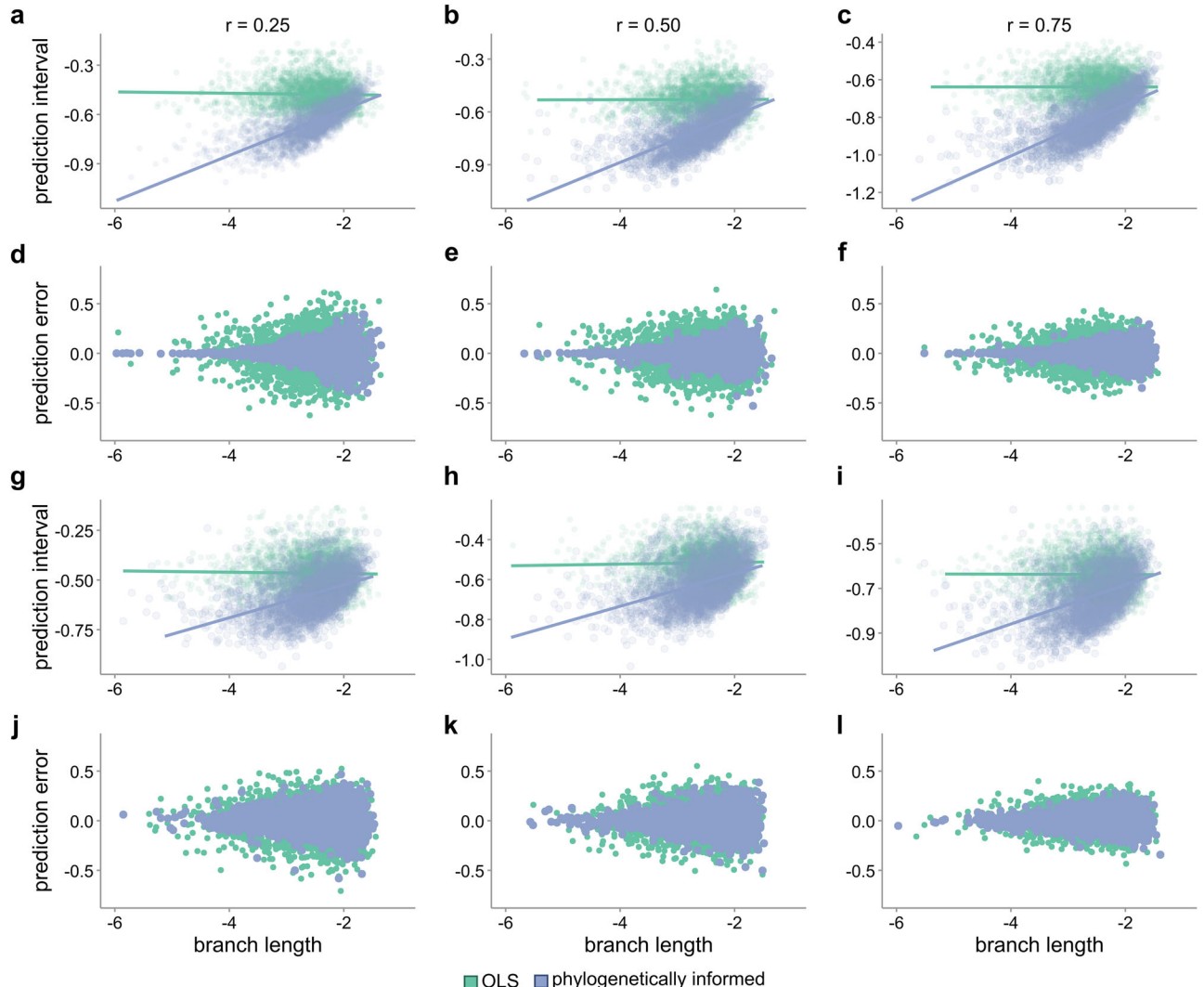

**Fig. 2 | Phylogenetically informed prediction intervals and error increase with terminal branch lengths. a–c** Prediction intervals (log$_{10}$-transformed) of phylogenetically informed predictions (blue points) increase with terminal branch lengths (log$_{10}$-transformed), whereas ordinary least squares (OLS) prediction intervals do not (green points), across set of 1000 simulated ultrametric trees (Fig. 1a) under three correlation coefficients ($r = 0.25$, 0.50, and 0.75). **d–f** Range in phylogenetically informed prediction errors (blue points) increase with terminal branch lengths (log$_{10}$-transformed) across set of ultrametric trees, whereas OLS-derived errors do not (green points). **g–i** Phylogenetically informed prediction intervals increase with terminal branch lengths of target extinct taxa across set of 1000 non-ultrametric trees (low extinction, see Fig. 1e). **j–l** Range in phylogenetically informed prediction errors increase with terminal branch lengths across set of non-ultrametric trees.

uncertainty surrounding our estimates is not only expected, but important. That is, phylogenetically informed predictions along with appropriately calculated prediction intervals (see Methods) are a more truthful representation of evolutionary reality than any single point estimate could provide.

**Predicting from biased data.** In the previous simulations, the target taxa were either randomly selected across the tree or were restricted to the extinct taxa. However, missing data in real-world applications are often non-randomly distributed across the tree of life[38,39], as we also find with our case studies (see Case study 1 below). We therefore ran two additional sets of simulations for the ultrametric trees and non-ultrametric trees with low extinction in which we restricted missing data to the lower quartile of the predicted trait (see Methods). As expected, we recovered a negative bias across all the simulations; however, phylogenetically informed predictions are less biased than the predictive equations (Fig. 3). Phylogenetically informed prediction performs about 1.6–4.9× better than predictive equations in ultrametric trees and about 1.3–3× better than predictive equations in non-

ultrametric trees with low extinction (Supplementary Data 12). Phylogenetically informed prediction is also more accurate than both predictive equations (estimated average error difference = 0.066–0.11 in ultrametric trees and 0.069–0.14 in non-ultrametric trees with low extinction, $p$-values < 0.0001; Supplementary Data 13).

These results further demonstrate the improved performance of phylogenetically informed prediction even in the case of non-random missing data, emphasising the necessity of accounting for shared ancestry when making predictions from real data.

**Case studies**

Our simulations reflect a standard for inferring unknown values where we have directly measured and known values for every trait of every species—both living and extinct. Each species is directly incorporated into the phylogenetic tree. However, in most scenarios—especially when seeking to infer characteristics of extinct taxa—we are unlikely to have such detailed information. Here, we use four case studies to highlight ways of overcoming commonly encountered issues faced by those reconstructing the unknown from real biological data: only

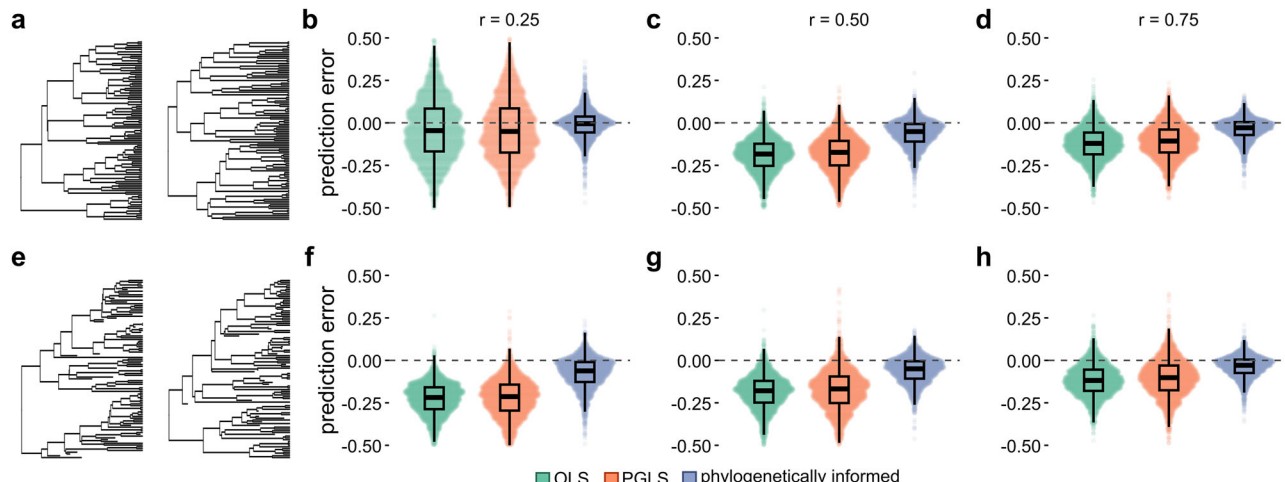

**Fig. 3 | Performance of phylogenetically informed prediction against predictive equations on biased data. a–d** Distributions of prediction errors (actual−predicted values) from a set of 1000 simulated ultrametric trees ($n = 100$ tips) under three correlation coefficients ($r = 0.25$, 0.5, and 0.75) with missing taxa exclusively from the lower quartile of the predicted trait. **e–h** Distributions of prediction errors from 1000 non-ultrametric trees ($n = 100$ tips) with low (**e–h**) extinction rates. Boxplot elements are as follows: centre line, median; box limits, first and third quartiles; whiskers, 1.5 × inter-quartile range; points, outliers. Colours represent the

three prediction methods (OLS: ordinary least squares predictive equations, green; PGLS: phylogenetic generalised least squares predictive equations, orange; phylogenetically informed predictions, blue). Tighter distributions near zero indicate more accurate predictions overall than those more skewed away from zero (measured by the variance in prediction errors, $\sigma^2$). Predictions are negatively skewed (larger than the actual values), but phylogenetically informed predictions are more accurate and less biased than other methods. Medians and variances of distributions can be found in Supplementary Data 12.

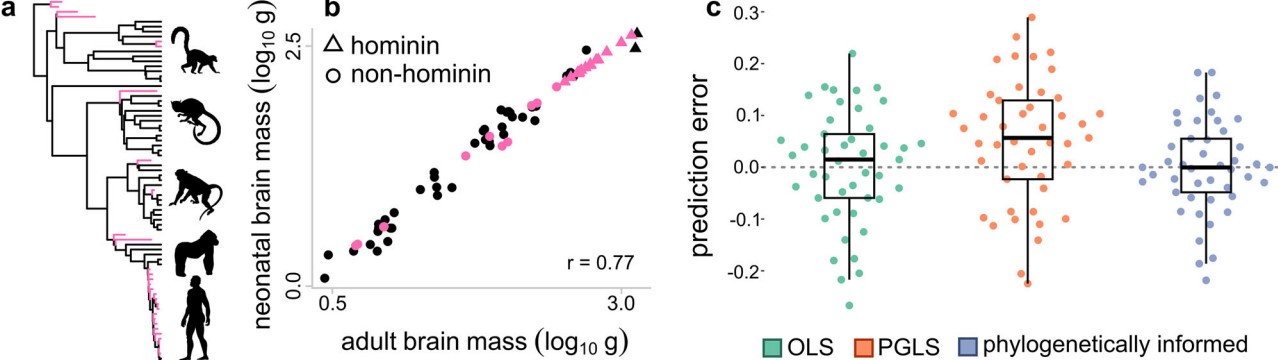

**Fig. 4 | Predicting neonatal brain size from adult brain size in primates.**
**a** Phylogenetic tree of primates including all taxa from which the predictive models are derived (black, $n = 254$) and all taxa for which imputed neonatal brain size (pink, $n = 30$). Silhouettes at the tips are not to scale, are purely for illustrative purposes, and are all obtained from phylopic.org. **b** A plot of the data used in the model overlaid with the predicted values from the phylogenetic inference model in pink. **c** Distributions of prediction errors (actual−predicted values) from our self-validation analysis ($n = 45$ taxa with known neonatal and adult brain sizes) using the

three different inference methods (OLS: ordinary least squares predictive equations, green; PGLS: phylogenetic generalised least squares predictive equations, orange; phylogenetically informed predictions, blue). Boxplots elements are as follows: centre line, median; box limits, first and third quartiles; whiskers, 1.5 × inter-quartile range; points, outliers. Tighter distributions near zero indicate more accurate predictions than distributions skewed away from zero $\sigma^2 = 0.012$ for OLS predictive equation, 0.015 for PGLS predictive equation, and 0.008 for phylogenetically informed prediction).

moderate phylogenetic signal (neonatal brain size in primates[40]), incomplete phylogenetic trees (bird body mass[23]), no branch length information (bush-cricket carrier frequency[41]), or data and model uncertainty (dinosaur neuron number[21]). In each case, we discuss the behaviour and interpretation of phylogenetically informed predictions in relation to true evolutionary scenarios.

**Case study 1: Phylogenetic signal and model performance.** Inferences about how neonatal primate brain size has evolved have fundamental implications for development, intelligence, and obstetrics[42–44]. Neonatal brain size is strongly linked to adult brain size, to the extent that adult female brain size has previously been used as a proxy for neonatal brain size[45]. The relationship between neonatal brain size and adult brain size estimated using PGLS across 44 extant primates[40] is significantly positive (y-intercept ($\alpha$) = −0.207, p-value for y-intercept ($p_\alpha$) = 0.009,

slope ($\beta$) = 0.89, p-value for slope ($p_\beta$) < 0.001), using a recently built and time-calibrated phylogenetic tree for primates[46] (see Supplementary Methods). The adjusted $R^2$ of the model is 0.923. There is only moderate phylogenetic signal in these data ($\lambda = 0.56$; here and throughout we use Pagel's $\lambda$[47] to refer to the strength of phylogenetic signal). Moderate or low phylogenetic signal has previously been used as a justification for using non-phylogenetic regressions to generate predictive models[23,48,49]. However, "limited phylogenetic dependence" is different from complete phylogenetic non-independence. It is better to incorporate some level of phylogenetic information than to ignore it completely.

We used phylogenetically informed prediction to reconstruct the neonatal brain size of 29 extinct species included in the phylogenetic tree (Fig. 4a, b). To assess model performance in the same way as our simulations, we used a self-validation approach to infer the neonatal brain size of each taxon in the model given its phylogenetic position

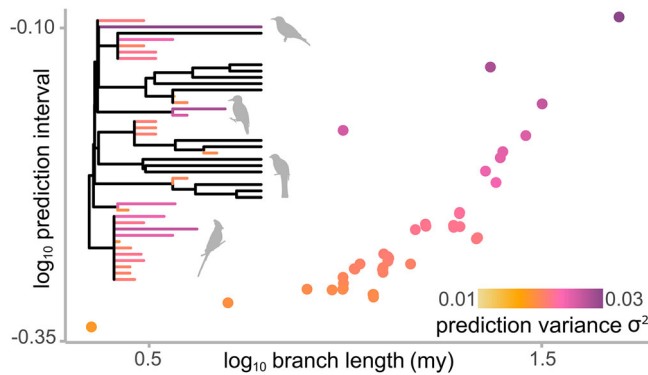

**Fig. 5 | Variance associated with phylogenetically informed prediction increases along branch lengths.** The scatter plot depicts the relationship between branch length and the variance associated with the predictions from our second case study. Inset, part of the phylogenetic tree ($n = 43$) with branches of extinct taxa (and corresponding points) coloured by the variance associated with their phylogenetically informed prediction for body mass. Silhouettes are not to scale.

(see Methods). We also calculated neonatal brain size for all extinct species using both the PGLS (parameters as above) and OLS predictive equations. As in our simulations on trees of similar structure (Fig. 1e), the performance of phylogenetically informed prediction far outstrips that of both predictive equations which both perform similarly (Fig. 4c).

Notably, PGLS predictive equations result in a strong, observable bias—with median prediction errors shifted away from zero (median = 0.06). This is compared to a much smaller bias observed using OLS predictive equations (median = 0.01), whereas the phylogenetically informed predictions are centred effectively on zero. That is, we observe a bias in both predictive equations that is much stronger in equations derived from PGLS regression (Box 1).

This case study represents a situation in which we have directly measurable trait data, a large sample of taxa, and where all species (both measured and unknown) are included within a robust phylogenetic framework. It is therefore as close as we are likely to get to our simulated scenarios using real data.

**Case study 2: Incomplete phylogenetic trees.** Whole papers and books have been published devoted to providing equations for predicting body size for various groups of animals[23,25,29]. Here, we predict body mass from humerus length for a total of 247 extant birds[23] matched to the Time Tree of Life[50,51]. This dataset has previously been used to predict the body masses of other extinct birds using a suite of non-phylogenetic predictive equations[22,52]. There is a strongly significant positive relationship between humerus length and body mass using both OLS ($\alpha = -1.08$, $p_\alpha < 0.001$, $\beta = 2.02$, $p_\beta < 0.001$, $R^2 = 0.93$) and PGLS models ($\alpha = -1.09$, $p_\alpha = 0.009$, $\beta = 2.07$, $p_\beta < 0.001$, $R^2 = 0.875$, $\lambda = 0.815$).

We obtained humerus measurements for a total of 41 fossil taxa[22], which were not included in the tree. In an ideal world, we would construct a tree that includes all taxa of interest; although, this is not always practical or even possible. However, rather than abandoning phylogenetic information altogether, it is possible to use the information we do have—taxonomic affiliation and, at least for fossils, temporal information—to maximise the accuracy of our inferences. This is possible in even the most drastic of circumstances where we lack a phylogenetic tree altogether: it is possible to construct phylogenetic frameworks purely from taxonomic information using standard tools for phylogenetic analysis[53]. Here, we have a robust phylogenetic tree available for living bird species[50,51] that we used as the basis for our phylogeny. For each species, we used taxonomic information and temporal

ranges to manually insert them into the tree. To do this, we identified the closest living relative in the tree, starting at the lowest possible taxonomic level (genus) and inserted the extinct taxon into the tree as a tip diverging either along the branch leading to its closest relatives or as a polytomy at the base of a clade (e.g., Fig. 5; see Methods and Supplementary Methods).

Such manual manipulation can impose structure on the topology where none exists. Furthermore, the greater our uncertainty regarding phylogenetic placement, the further back in the tree the taxon will be positioned (e.g., the base of a family, order, etc.). All else being equal, the more taxa we insert, the more likely it is that some branch lengths in the tree will be longer than they should be. As we are studying an evolving system, a longer branch has more evolutionary potential (in terms of change per unit time; Fig. 2). We find that prediction intervals increase alongside increasing branch length (Fig. 5, $r = 0.81$, p < 0.001), in line with our simulation results across trees with and without extinction (Fig. 2, Supplementary Fig. 2). Whilst phylogenetically informed prediction may provide answers with wider prediction intervals, they are more likely to reflect true evolutionary outcomes. As noted by Garland and Ives[11] (pg. 357): "*what phylogeny taketh away, phylogeny giveth back.*".

**Case study 3: No branch-length information.** In our previous case study, we highlighted the importance of phylogenetic structure, noting that prediction intervals increase with branch length. However, it is often the case—especially in cladistic analyses or in taxonomy-based trees—that we have no branch length information. Nonetheless, it is still possible to use such datasets for phylogenetically informed prediction (Supplementary Fig. 3, Fig. 6).

Here, we use a published dataset and tree[41] for 94 species of bush-crickets (or katydids) to study the relationship between the size of a specialised structure on the wing known as the stridulatory file and carrier frequency[41,54]. However, the tree inferred using parsimony methods has no branch lengths. There are various approaches for assigning branch lengths to a phylogenetic tree for use in comparative analysis[55,56], though largely this choice is arbitrary with regards to the inferences made from regression models[55]. Here, we use the common approach of setting equal branch lengths[21,57,58] (i.e., all branch lengths = 1, Fig. 6a), and insert 18 species as polytomies into the tree based on family and sub-family membership (Supplementary Methods). We then used all three methods to predict the call frequency for these taxa.

The estimated non-phylogenetic relationship between file length and call frequency is equivalent to that recovered previously[41] ($\alpha = 1.66$, $p_\alpha < 0.001$, $\beta = -1.04$, $p_\beta < 0.001$, $R^2 = 0.66$). The phylogenetic relationship is very similar ($\alpha = 1.62$, $p_\alpha < 0.001$, $\beta = -0.97$, $p_\beta < 0.001$, $R^2 = 0.53$), but there is strong phylogenetic signal in this dataset (Pagel's $\lambda = 0.821$). When we compare the pairwise differences among the predictions made by the three different methods (Fig. 6b), it becomes apparent that phylogenetically informed prediction produces values that span a greater range of variation in comparison to the predictive equations. Predictions made from OLS and PGLS regression equations are both drawn from a single slope and thus differ from each other only negligibly (Fig. 6b). On the other hand, phylogenetically informed prediction tends to differ from both OLS and PGLS predictions—although, the effect is less pronounced in PGLS (Fig. 6b). The reason why phylogenetically informed predictions appear more different is because we are not forcing each point to fall along a single line. This is a closely related but distinct point from our simulations and the previous case study (Figs. 2 and 5). In those instances, we demonstrated that the prediction intervals of a single estimate increase with branch length, whereas here we show that the variation among estimates increases. In both scenarios, we can attribute the variance to the fact that we are examining an imperfect, evolving relationship.

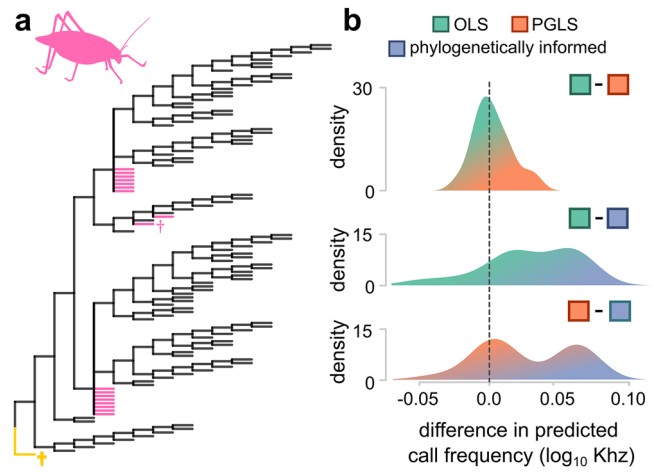

**Fig. 6 | Predictions drawn from a single equation are missing a dimension of variation. a** Predictions of call frequency were made on a tree with no branch-length information, where all branches were fixed to one and all taxa were inserted as polytomies (total $n = 112$). Extinct taxa are marked with †. *Archaboilus*, marked in yellow, belongs to a different family to all other species in the tree. Species with no call frequency data are marked with pink ($n = 18$). **b** Distributions of pairwise differences between each of the three methods being compared (OLS: ordinary least squares predictive equations, green; PGLS: phylogenetic generalised least squares predictive equations, orange; phylogenetically informed predictions, blue), demonstrating hidden variation in the estimates that cannot be observed using simple equations.

For the most part, prediction accuracy will be largely unaffected—but, depending on the units of branch length measurement along with those of the traits being studied, prediction variance can affect interpretations[16,56] and drastically affect prediction intervals (Fig. 2). For example, where we have poorly estimated variance (e.g., in trees with equal branch lengths or in trees inferred exclusively from taxonomy), it would be difficult to draw meaningful conclusions from any tests relying on robust estimates of prediction variance (e.g., outlier tests[16]). It will therefore always be preferable to include dating information from fossils or molecular clocks where possible, to minimise variance inflation or reduction from incorporating incorrect information.

The carrier frequency is one of just a few characteristics that needs to be known to reconstruct the sound of a stridulating insect[26,41]. It is certainly an exciting prospect to reconstruct the sounds of the past, yet previous attempts to do so used predictive equations that lack evolutionary information[26,59]. Here, we reconstruct the calling frequency in two extinct species (Fig. 6). For *Pseudotettigonia amoena*, we reconstruct a mating call frequency of 9.97 kHz ($\log_{10}$ prediction = 0.99, 95% PI[0.568,1.430]) compared with previous estimates of 10.5 kHz derived from the same morphology[41]. For *Archaboilus*, we estimate a value of 4.78 kHz ($\log_{10}$ prediction = 0.679, 95% PI[0.321,1.037]) compared with 4.99 kHz[41]. However, *Archaboilus* is exemplary of phylogenetic extrapolation, a problematic scenario often encountered when reconstructing values for extinct taxa (Fig. 1i–l, see also Fig. 4 in which two species fall outside the euprimate clade). In the case of *Archaboilus*, this species is not a bush-cricket, rather belonging to an extinct family (Haglidae). This problem will be exacerbated by the lack of branch length information—taxa that fall outside the range of extant phylogenetic diversity are more likely to have long branches and therefore greater uncertainty in predictions (Figs. 2 and 5).

**Case study 4: Data and model choice.** Our previous two case studies exemplify situations in which we have uncertain or incomplete phylogenetic information. However, what about the data that goes into the analysis in the first place? Here, we present a case study seeking to

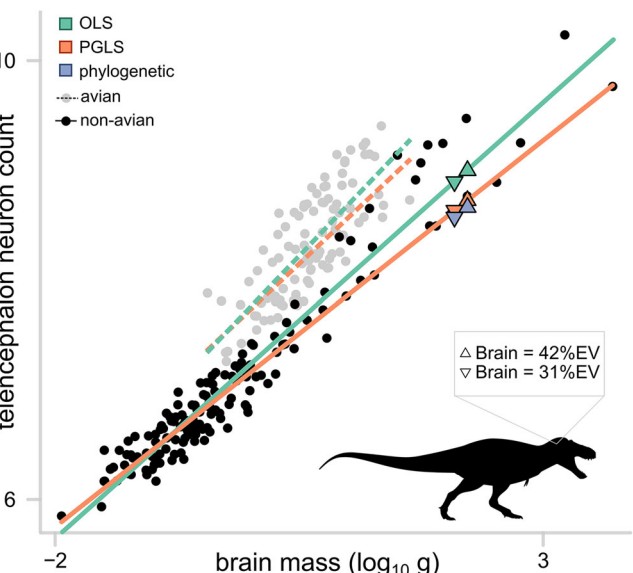

**Fig. 7 | Predicting telencephalon neuron number from brain mass for *Tyrannosaurus rex*.** The relationship between telencephalon neuron number and brain mass (both logged) was estimated across a sample of 202 sauropsid species using both ordinary least squares (OLS, green) and phylogenetic generalised least squares (PGLS, orange) regression models. We then predicted the neuron count for *T. rex* using both model equations (see text, green and orange triangles) as well as phylogenetic inference (blue triangles), assuming the relationship in *T. rex* aligns with non-avian sauropsids. This was done using two alternative brain size measurements: firstly, if the brain takes up 42% of the braincase (triangle pointing upwards) and secondly, if the brain takes up 31% of the braincase (triangle pointing downwards). Altering the ratio obtains qualitatively similar conclusions, but the inferred values simply 'slide' up or down the regression line. The two measures were calculated as an average across 3 different specimens, but results are qualitatively identical when each specimen is considered separately. A separate slope was estimated for avian taxa (grey points, dashed lines).

reconstruct the number of neurons in the brains of an extinct dinosaur from brain size to highlight this potential issue.

Recent research suggests that large theropod dinosaurs, such as *Tyrannosaurus rex*, were likely to have been highly intelligent[24] based on inferred telencephalic neuron number derived from OLS predictive equations. A recently published critique of the original research[21] challenges the idea that large-bodied theropod dinosaurs had excessively large neuron counts using PGLS predictive equations. This case study therefore represents a unique opportunity in which we can assess the impact of phylogenetically informed prediction on biological conclusions.

Here, we use a dataset of 254 sauropsid species[21] matched to the Time Tree of Life[51], cropped to only include species in the dataset. For *T. rex*, we use the brain size data provided by Caspar et al.[21]. We then grafted *T. rex* onto the tree using recent estimates of dinosaur divergence times[60] (see Supplementary Methods). Previous researchers have noted the difference in brain-body size relationships and neuron densities between birds and other non-avian sauropsids[21,24,61]. This scenario therefore also represents an opportunity to demonstrate the use of multiple regression models to predict unknown trait values: we are not limited to the simple bivariate relationships demonstrated thus far. In this case, we used a phylogenetic analysis of covariance to estimate a separate slope and intercept in the relationship between neuron count and brain mass for avian and non-avian sauropsids, in line with both Caspar et al.[21] and Herculano-Houzel[24] (Fig. 7).

Our phylogenetically informed prediction results largely align with the main conclusions drawn by Caspar et al.[21] (Fig. 7). It is unlikely that *T. rex* had neuron numbers indicative of any spectacular

intelligence or cognition. The OLS regression equation ($\log_{10} neuron\ count = 7.77 + 0.93[\log_{10}brain] - -0.52[\text{non-avian}] - 0.13[\text{non-avian} : \log_{10}brain]$) predicts *T. rex* to have had ~771 million neurons, although there is a great deal of uncertainty in this estimate ($\log_{10}$ prediction = 8.885, 95% PI[8.378,9.391]). By virtue of the shallower slope, the PGLS regression equation ($\log_{10} neuron\ count = 7.74 + 0.84[\log_{10}brain] - 0.56[\text{non-avian}] - 0.13[\text{non-avian} : \log_{10}brain]$) produces a much smaller estimate of ~422 million ($\log_{10}$ prediction = 8.626, 95% PI[8.339,8.912]). When we account for the phylogenetic position using phylogenetically informed prediction, we see yet again a smaller estimate of ~378 million with wider prediction intervals ($\log_{10}$ prediction = 8.564, 95% PI[8.005,9.112]). The increase in the width of the prediction intervals reflects the fact that *T. rex* falls along a very long branch outside of modern birds.

However, this is just one part of the story. The equations above are from a model using brain masses converted from volumes assuming that the brain takes up 31% of the endocranial volume (%EV). However, %EV is variable among taxa ranging from 30% in the tuatara[62], 31–42% in crocodilian species[21], to up to 100% in modern birds, and people have used various values to calculate dinosaur brain size. Changing the %EV in our calculations simply 'slides' our predictions up and down the relevant regression slope (e.g., Fig. 7). However, recent studies suggest that brain development in non-avian dinosaurs, including theropod species like *T. rex* is highly divergent compared to both crocodilians and modern birds[63]. Therefore, we cannot currently say with any certainty what value (or range of values) is most likely to represent the percentage of the endocranial volume taken up by the brains of dinosaurs.

The most striking difference in the estimate of neuron number between the two original papers[21,24] comes not from the methods, but rather from the taxonomic treatment of the missing species. That is, whether non-avian theropod dinosaurs like the *T. rex* are estimated using an avian or non-avian scaling relationship. Unsurprisingly, when *T. rex* is predicted according to the steeper scaling relationships observed in birds (dashed lines, Fig. 7), the inferred neuron number is much larger using all methods[21,24]. However, *T. rex* is an extreme case of both data and phylogenetic extrapolation under the avian treatment, falling outside the range of observed data for the avian group. In terms of relative brain size, *T. rex* falls somewhere in between modern birds and non-avian sauropsids[21], and most importantly, falls at the boundaries of extant variation (Fig. 7). Taken together, this means that however we treat this species, our estimate is fraught with uncertainty −uncertainty that is, at least, partially reflected in the extraordinarily wide prediction intervals for all methods, which range across orders of magnitude.

Using statistically appropriate methods provides us with a more accurate depiction of extinct taxa, but it can only get us so far. The ways in which we define brain size and evolutionary relationships of an animal can produce predictive models that drastically differ (Fig. 7). In the absence of additional data or analyses, the best way to treat dinosaurs−and particularly theropod dinosaurs like tyrannosaurids with unusually large brain regions[64]−remains unclear. We agree with the conclusions outlined by Caspar et al.[21]: making inferences about complex biological characteristics such as cognition and behaviour are likely to remain difficult and are better considered within multidimensional integrative empirical frameworks.

### Recommendations and future directions

It may be obvious that OLS equations are inadequate for making predictions in comparative biological studies because they do not model evolutionary history. Why PGLS equations are insufficient is more subtle. It is true that PGLS equations model the evolutionary structure of comparative data but using them to predict values for a taxon assumes it is "phylogenetically average". That is, its location on the tree is unknown and it is implicitly placed at the root. Phylogenetically informed predictions, on the other hand, are much more powerful because the taxon's location in the tree is used as additional information.

In Box 2, we provide a primer for making phylogenetically informed predictions to infer trait values. Although phylogenetically informed prediction is, on average, much more accurate and precise, it is no holy grail. No method is. And, as with all statistical models, assumptions, context, and a priori expectations should guide the design and interpretation for prediction in comparative analysis. Yet even in worst-case scenarios, where there is no phylogenetic signal in the traits of interest, phylogenetically informed prediction will do no worse than other predictive methods. Edge cases also include phylogenetic extrapolation (Figs. 1 and 6) where predictions are made outside of a comparative framework (i.e., prediction of outgroups). More commonly, taxa are related in ways that substantially improve our ability to make predictions in evolutionary and comparative studies. This framework is especially important for long-isolated lineages where uncertainty accumulates with terminal branch lengths (Figs. 2 and 5, Supplementary Fig. 2). The case studies we highlight showcase specific and common problems researchers are likely to face when inferring unknown values, ranging from phylogenetic uncertainty to model construction. Researchers must consider all aspects of the system they are interested in studying before embarking on attempts to make predictions.

Reconstructing trait values is a fundamental and ubiquitously important tool across biological sciences. Examples include predictions and retrodictions to estimate unknown values for traits[6,17,65,66] or to test for exceptional evolutionary singularities[16], as well as imputations, and extrapolation. Our results show that evolutionary history substantially improves prediction of continuous traits in evolving systems, which in turn, clarifies and constrains evolutionary hypotheses.

## Methods
### Simulation
To simulate trees, we used a version of the Bellman–Harris model[67] from the R[68] package TreeSimGM[69], which samples waiting times until speciation and extinction from a specified distribution for a given number of extant taxa (sim.taxa function) or duration of time (sim.age function). The model allows for symmetric (cladogenic) and asymmetric (budding) modes of speciation. We first simulated 1000 ultrametric trees (sim.taxa, *n* = 100 taxa), sampling speciation times from an exponential distribution with a rate of 1 without extinction (extinction rate of 0) while allowing for asymmetric speciation (Fig. 1a). The 1000 ultrametric trees had varying tree shapes, including different degrees of balance and stemminess. Balance (and imbalance) refers to the degree at which the tree subsets are symmetrical in length or size. Stemminess refers to the number of speciation events concentrated at the tips (high stemminess) or toward the root (low stemminess). This protocol resulted in trees reflective of those made from real datasets (see case studies). We also simulated completely balanced and imbalanced ultrametric trees, representing both extremes in tree shape. To simulate the completely balanced tree (Supplementary Fig. 1a), we allowed the waiting times to speciation to be shorter, on average, than extinction (sim.taxa, *n* = 128 taxa, exponential distribution: speciation and extinction rates of 0.4 and 0.5, respectively) and only allowed for symmetric speciation. For the imbalanced tree (Supplementary Fig. 1e), we set the waiting times to extinction to be as long as the age of the tree while allowing for asymmetric speciation, creating a highly imbalanced ultrametric tree (sim.age, age = 2, exponential distribution: speciation and extinction rates of 0.02 and 2, respectively, resulting in 100 taxa).

To assess the performance of predicting extinct taxa, we simulated 1000 non-ultrametric trees with low extinction rates (sim.taxa,

## BOX 2
# A roadmap to successful phylogenetic prediction

**1.** *Consider the data*: Understanding the data is paramount for successful predictive modelling. Does the trait of interest vary with multiple traits? Taxonomic group? Time? It is possible to incorporate multiple variables into phylogenetic prediction (e.g., Case study 4). The power phylogeny plays to improve predictions increases with noisy data because there is more residual variance to explain. Does it have strong phylogenetic signal? Strength of phylogenetic signal will be especially important for univariate predictions. A strong relationship (Fig. A) will provide better predictions than a weak or highly variable one (Fig. B), but the phylogeny will play a smaller role.

**2.** *Consider the tree*: A tree encompassing the taxonomic diversity of not only the species in the data but of those we want to predict is important. The accuracy of predictions will diminish as taxa become more distantly related to those from which predictions are derived (e.g. compare pink taxa in Fig. C). *Phylogenetic extrapolation*—predicting species that fall outside the phylogenetic diversity of the group altogether (e.g., Case studies 1 and 3) should be avoided. A tree with branch lengths (temporal, molecular, etc.) is desirable, but not essential (Fig. 6; Supplementary Fig. 3).

**3.** *Decide on an approach*: We use maximum-likelihood PGLS models for phylogenetic prediction. However, there are many approaches to incorporate phylogenetic information directly into predictive modelling—including alternative statistical frameworks such as Restricted Maximum Likelihood (REML) and Bayesian approaches[6]. In our analyses, we use trees with branch lengths transformed both by Pagel's λ and κ[47], but it is possible to incorporate various types of tree transforms into predictive models—so long as these act homogenously across all branches (e.g., Ornstein-Uhlenbeck[72]). There are various packages, functions, and programs for phylogenetic prediction (of all types, including imputations from only a single trait, eigenvector, structural equation, and missing forest approaches not discussed further here). Here we provide a non-exhaustive list: R:Phylopars[90]; PhyloPars[91]; BayesTraits[6]; R:phytools[92]; R:MCMCglmm[14]; R:picante[93]; BayesModelS[94]; R:missForest[95]; R:MPSEM[96]; R:VIM[97]; R:mice[98]; R:BHPMF[99]; R:phylosem[100].

**4.** *Interpret the results*: No prediction method is without error. Phylogenetic prediction explicitly reveals the expected variance around an estimate. Whilst it might sometimes seem preferable to obtain a point value, such variance represents evolutionary reality and can be quantified using prediction intervals. Prediction intervals increase not only with time, but with increasing phylogenetic distance (Figs. 2, 5, Supplementary Fig. 2). It is important to remember that predicting is not just about accuracy—but also honesty.

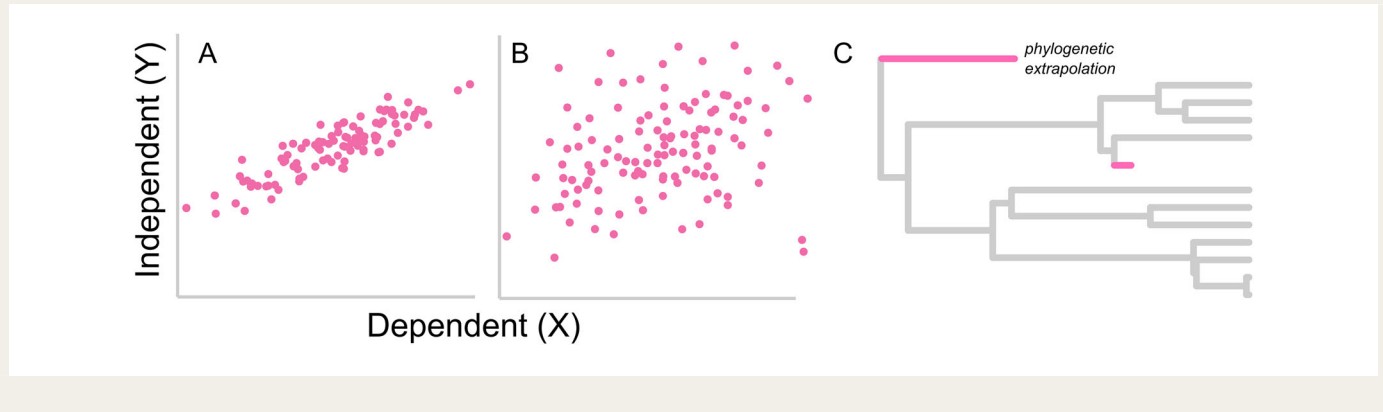

n = 100 extant taxa; Fig. 1e), sampling speciation and extinction times from an exponential distribution with rates of 1 and 0.2, respectively. We also simulated 1000 non-ultrametric trees with high extinction rates (sim.taxa, n = 100 extant taxa, exponential distribution: speciation and extinction rates of 1 and 0.9, respectively; Fig. 1i). In addition, we simulated an extreme scenario in which extinction was constant, creating a pectinate or ladder-like tree (sim.age, age = 50, exponential distribution: speciation and extinction rates of 0.5 and 1, respectively, resulting in 100 taxa total; Supplementary Fig. 1i). We simulated all trees to end with 100 extant taxa (except the completely balanced tree, which had 128 extant taxa). The simulations with extinction yielded trees with more than 100 total taxa. To make these simulations consistent with the ultrametric trees, we randomly down sampled the number of total taxa to 100. Given the demonstrated relationship between branch length and prediction variance (Fig. 2 and Supplementary Fig. 2), we further standardised all ultrametric and non-ultrametric trees to a length of 1.

For each tree, we simulated two phylogenetically structured continuously correlated traits under a multivariate Brownian motion model (expected amount of change is distributed normally with mean

0 and variance proportional to the branch lengths of the simulated trees), using the function sim.corrs in the R package phytools[10]. We simulated three sets of datasets with correlation coefficients of 0.25, 0.5, and 0.75. Because the characters were simulated along phylogenies, they are expected to covary according to their respective phylogenies, and our estimation of phylogenetic signal is high with an average λ = 1 in all the datasets, where λ refers to Pagel's lambda[47], which varies between 0 = no phylogenetic signal and 1 = high phylogenetic signal).

For the ultrametric trees, we randomly removed the dependent trait for 10 taxa (10% of the tips) and stored the actual values. We removed all the dependent traits of the extinct taxa in the non-ultrametric trees. Then, we predicted the values of the removed dependent traits using a maximum likelihood PGLS regression model that accounts for phylogenetic non-independence in the prediction[70]. All of our phylogenetically informed predictions are performed in R[68], and we provide the source code for the full method in Supplementary Code 1. We estimated phylogenetic signal using Pagel's λ for each simulated dataset to account for phylogenetic non-independence. We compared the maximum likelihood estimates of the phylogenetically

informed predictions to those estimated using equations derived from OLS and PGLS regressions. We fit OLS and PGLS regression models to the data (excluding taxa with removed dependent traits) using the functions lm and pgls in the base and caper[71] R packages, respectively. To determine the accuracy of the three prediction methods (phylogenetically informed, OLS, and PGLS), we took the difference in the estimated prediction from the actual value of each taxon. We calculated the prediction error (actual−predicted) of all predicted taxa for every tree and for each method. We compared the distribution of prediction errors among the three methods for each tree type and dataset (Figs. 1 and 3, Supplementary Figs. 1 and 3). As a measure of overall predictive performance, we compared the fold decrease in the variance of prediction errors between the predictive equations and phylogenetically informed prediction (overall performance = $\sigma^2$ in OLS or PGLS predictive equation errors divided by $\sigma^2$ in phylogenetically informed prediction errors). Medians and variances in prediction errors for each method, tree type, and dataset can be found in our Supplementary Data 1, 4, 6, 8, 10, and 12. The analyses on the sets of ultrametric trees and non-ultrametric trees with low extinction were repeated for different tree sizes (n = 50, 250, and 500 taxa; Supplementary Data 4).

To compare the accuracy of phylogenetically informed predictions against OLS and PGLS predictive equations, we calculated the difference in the absolute prediction errors between each predictive equation and the phylogenetically informed predictions (error difference = absolute prediction error for OLS or PGLS equations−absolute prediction error for phylogenetically informed prediction). We took the absolute values of the prediction errors because the errors can be positive or negative. Prediction error differences were calculated separately for OLS and PGLS equations, both compared against the phylogenetically informed prediction error. If the prediction error difference is positive, then the predictive equation has the larger error and is less accurate (i.e., phylogenetically informed prediction is more accurate). The predictive equations are more accurate if the error difference is negative. We then calculated the median error difference for every simulated tree, resulting in 1000 differences in prediction error (3,000 total across the three correlations: r = 0.25, 0.5, and 0.75) each for OLS and PGLS equations. To test if phylogenetically informed predictions are more accurate than OLS and PGLS equations, we ran intercept-only models on the median prediction error differences against a mean of 0−amounting to a one-sample t test, assessing whether the median error differences are significantly different from 0. The results of these analyses can be found in our Supplementary Data 2, 5, 7, 9, 11, and 13.

A common strategy for predicting taxa without known phylogenetic information (e.g., uncertain phylogenetic position, no branch lengths, etc.) is to force all branch lengths to the same length. To assess the impact this strategy has on the accuracy of phylogenetically informed predictions, we took the set of ultrametric and non-ultrametric trees with low extinction (Fig. 1a,e) and scaled all branches to a length of 1 using the κ parameter[47] (κ = 0). It is also possible to incorporate different types of tree transforms into phylogenetically informed prediction, as long as all branches are scaled homogenously (e.g., Ornstein-Uhlenbeck[72]). Trees were scaled using the rescale function in the R package phytools[10]. Retaining the original datasets, we then used the maximum likelihood phylogenetically informed prediction method described above and recalculated the prediction errors.

To test the effects of tree shape on prediction accuracy, we calculated the following three metrics for the distributions of simulated ultrametric and non-ultrametric trees: Colless's imbalance metric[73], Sackin's imbalance metric[74], and Rohlf et al. (1990)'s stemminess metric[75]. The imbalance metrics were calculated using the collessI and sackinI functions in the package treebalance[36]. The stemminess metric was calculated using R code from Humphreys et al.[76] (Supplementary

Code 1). We ran linear models testing the effects of each tree shape metric on the absolute values of the prediction errors for the phylogenetically informed predictions and PGLS predictive equations (Supplementary Data 3).

Along with prediction error, we also calculated phylogenetically informed prediction intervals for our simulations and case studies, according to the GLS approach by Garland and Ives[11]. Prediction intervals provide a range of values that we can estimate under a certain probability (e.g., 95%) will encompass the new observation. These intervals are like confidence intervals, except they include both the underlying model variance and the expected variance around the prediction. Unlike prediction intervals derived from OLS regression, phylogenetically informed prediction intervals explicitly incorporate phylogenetic information of the target taxon into the calculation of the prediction variance. Phylogenetically informed prediction intervals are, therefore, expected to increase with the terminal branch length of the target taxon, according to Brownian motion, whereas those from OLS regression will not. Prediction intervals in a PGLS regression framework can also be calculated assuming the target taxon is attached to the root of the tree and has a branch length equal to the average length from the root to each tip ("generic" prediction intervals in Garland and Ives[11]). We calculated these generic PGLS prediction intervals (also referred to here as 'prediction intervals from PGLS predictive equations'), along with the OLS and phylogenetically informed prediction intervals and assessed how they varied with the terminal branch lengths of the target taxa.

In order to facilitate comparison across approaches, we derive all prediction intervals based on variances calculated using a Student's t-distribution (α = 0.05). This is in line with the base statistics packages that are commonly used to perform non-phylogenetic OLS regression analysis[68]. We note that in a phylogenetic context, it may be more intuitive to derive prediction intervals using variances calculated using z-distribution[77] but this makes no qualitative difference to any of the conclusions we draw from our results, especially at larger sample sizes (n > 30 taxa).

## Real datasets and trees

Our first case study set out to predict neonatal brain size from adult brain size in primates. We use a published brain size dataset for 44 extant primates[40] and added values for *Homo neanderthalensis* from additional sources[78,79]. All taxa are included in the recently published meta-tree analysis of Euarchonta[46]; we use a dated sample of 100 of the most parsimonious trees[80] (Supplementary Methods). For our prediction analyses, we used the median tree from this sample, calculated using the Kendall-Colijn metric[81]. We obtained adult brain size for an additional 29 primate species included in this tree (Fig. 4a), including 18 hominins[79] and 11 other species throughout the order[82–87]. These 29 species have no data on neonatal brain size and were therefore the species for which we predicted unknown values (Fig. 4b).

Our second case study set out to predict avian body mass from humerus length. We use a published dataset of 318 extant birds with body mass and humerus length[23]. Of these, we matched 247 species to the Time Tree of Life[51] (downloaded in February 2024[88]). We used a published dataset of humerus lengths for 41 extinct birds[22] as the species for which we predicted unknown values. However, the Time Tree of Life contains only extant taxa−and so in this case, none of our predicted species were found in the tree. For each of the species, we therefore identified the closest living relatives in the full time tree based on taxonomic information and manually inserted a branch as a sister taxon or into a polytomy at the base of the relevant clade. The date of divergence was set as the first appearance date in the fossil record and extended the branch to its last appearance date. For example, *Plesiocathartes kelleri* is identified as belonging to the order Leptosomiformes in the Paleobiology Database[89]. We therefore placed this *P. kelleri* as a sister taxon to an anchor species, *Leptosomus discolor*,

diverging at the maximum age of the clade and extended the branch to the minimum age of the fossil as reported in the original dataset[22]. A full description of the placements for all 41 taxa can be found in the Supplementary Information (Supplementary Methods).

Our third case study set out to predict the mating call carrier frequency of bush-crickets (katydids). We use a published dataset comprising wing measurements and carrier frequencies for 94 species along with a corresponding phylogenetic tree[41]. The paper from which the data and tree was obtained also provided a set of wing measurements for 18 species (living, museum, and extinct) that were not included in the phylogenetic tree. As with the second case study, we inserted each of the 18 species into the tree based on taxonomic information (see Supplementary Methods).

Our final case study set out to predict the telencephalic neuron number of the extinct theropod dinosaur, *T. rex*. To do this, we used a published dataset of 260 sauropsids[21] which represents a recently updated version of another recent dataset[24]. These previous datasets either used a composite tree by splicing together various group-level phylogenies[21]—or no tree at all[24]. We simplify the procedure—and avoid complications associated with mis-matched node ages in the phylogenetic tree by using the Time Tree of Life, which includes 254 species in the dataset. We then grafted *T. rex* into this phylogeny based on the dates inferred in a recently published analysis of dinosaurian divergence dating and phylogeny[60] (see Supplementary Methods).

The full datasets and corresponding phylogenetic trees for each of the four case studies can be found in our Supplementary Information (Supplementary Data 14).

### Case study predictions
For each case study, we make predictions for all three approaches as described for our simulations. We fit OLS and PGLS regression models to the data using the functions lm and pgls in the base[68] and caper[71] R packages, respectively, and estimate values for missing taxa using predictive equations derived from both models. We then predicted the values for missing taxa using a PGLS regression model that simultaneously accounts for phylogenetic non-independence in the prediction[70]—we did this using a custom R script for which we provide the source code in Supplementary Code 1. In both PGLS models we estimate phylogenetic signal using Pagel's $\lambda$[47].

For the first case study (neonatal brain size), we conducted an additional set of self-prediction analyses to validate the model. To do this, we ran the PGLS model on the full dataset but removed the residual value for each species (one at a time) and adjusted the phylogenetic weighting as appropriate. This allowed us to obtain predictions for all species included in the model. As with all our analyses, this was performed in R by way of a modified version of the function used in our phylogenetically informed predictions[70]—the full code is provided in Supplementary Code 1.

### Reporting summary
Further information on research design is available in the Nature Portfolio Reporting Summary linked to this article.

## Data availability
All datasets analysed during the current study are included in this published article (and its supplementary information files). The full datasets and corresponding phylogenetic trees for each of the four case studies can be found in the Supplementary Information (Supplementary Data 14) and were obtained from the following sources: Case Study 1—Wisniewski et al. (2022) *Proc. R. Soc. B.*[46], Gómez-Robles et al. (2024) *Nat. Ecol. Evol.*[40]; Case Study 2—Hedges et al. (2015) *Mol. Biol. Evol.*[51], Field et al. (2013) *PLOS* ONE.[23], Crouch et al. (2019) *Proc. R. Soc. B.*[22]; Case Study 3—Montealegre-Z et al. (2017) *J. Evo. Bio.*[41]; Case

Study 4—Caspar et al. (2024) *The Anat. Rec.*[21], Hedges et al. (2015) *Mol. Biol. Evol.*[51]

All datasets generated for our simulation analyses can be created using code provided in the Supplementary Information (see Code Availability).

## Code availability
All analyses were performed in openly available software as cited. We provide the full R code and associated statistical results used to generate the simulations as supplementary files. We also provide additional custom R code used for the imputation aspect of this research. All code is provided in Supplementary Code 1.

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

## Acknowledgements

The authors thank Kevin Surya, along with the Montana State University Deep Time Biology lab and University of Reading Evolutionary Biology Group, for helpful discussions and constructive comments on the analyses. This work was supported by a Leverhulme Trust Leadership Award (RL-2019-012) to CV All silhouettes in Figs. 4–7 were obtained from Phylopic (https://www.phylopic.org/) and are dedicated to the public domain.

## Author contributions

All authors jointly conceived and designed the study, interpreted the results, and prepared the manuscript. JDG and JB implemented the analyses, and CV and CLO supervised the work.

## Competing interests

The authors declare no competing interests.
