## [Transparent Peer Review file · Nature Communications]

Phylogenetically Informed Predictions Outperform Predictive Equations

Corresponding Author: Professor Chris Venditti

Version 0:

Reviewer comments:

Reviewer #1

(Remarks to the Author)

Good paper in general, but hard to follow. I'll resist saying "hard to see the forest for the trees" in some places. Needs some summary statistical tables.

Title and elsewhere: I don't think Phylogenetically-Informed needs a hyphen because the adjective ends in lly.

Abstract: "simulations" should be used here so the readers has some idea of what was done

Line

21 simply? hahaha

25 I am confused because PGLS can be used to make "phylogenetically informed" predictions. What's the distinction?

27 finish the thought ... by use of what?

28 correlated with what? do you mean two traits being correlated or do you mean that a given trait (or the residuals) possess phylogenetic signal (sensu Blomberg et al. 2003)?

32-33 absolutely

35 spelling?

55 I think you mean predict the value of a single trait, e.g., body size, as opposed to, say, predicting a Y (e.g., brain size) from an X variable (e.g., body size). Please clarify.

In general, this prose might be made more tangible and relatable with some examples sprinkled in.

64 Good point, and others could be cited from the allometry world.

65-66 confusing terminology/prose. We are talking about predictive equations (or procedures) in both cases.

69-70 Again, confusing, you are slamming PGLS predictive procedures or exalting them?

83 "extant trees" sound strange. Maybe "trees with only extant taxa"

90 Only used ultrametric? I was expecting some paleo relevance, which would use trees with some branches terminating before the present.

Oh, OK, I see it's down below.

94 Multivariate? Technically it is also bivariate, so I'd just say that.

112 Where are the stats shown? Refer to a table?

137-140 This is a cool result.

187 Same as setting them all to unity, right? Might say this as well. This is what you say on 285.

188 Well, setting branch lengths to anything IS an assumption for the subsequent analyses!!!

213 Avoid starting sentences with "There have been" because (1) it makes "there" the subject of the sentence and (2) is unnecessarily wordy. Try something like "Inferences about how neonatal primate brain size has evolved ..."
Please check for other such instances.

285 Some other approaches for coming up with arbitrary "starter" branch lengths are also common and should at least be mentioned:

Grafen, A. 1989. The phylogenetic regression. *Philosophical Transactions of the Royal Society of London. Series B, Biological Sciences* 326:119–157.

Garland, Jr., T., P. H. Harvey, and A. R. Ives. 1992. Procedures for the analysis of comparative data using phylogenetically independent contrasts. *Systematic Biology* 41:18–32.

289 These lambda, alpha and other symbols that are Greek to me come out of nowhere with no explanation. Please explain. Are we talking about Pagel's lambda? What about OU or ACDC transforms, as in Blomberg et al. 2003?

289 Also, I may have missed it, but what software was used to do all of this?

397 comparative biological studies

398-401 OK, now I think I get the distinction you are making between PGLS (which I have always considered "phylogenetically informed" because they are!) and what you are calling "phylogenetically informed" here. This is not great terminology, and whatever your distinction should be made clear in the Abstract and Introduction.

Box 1 Need to use "PGLS" here

Need to label the three clearly as OLS, PGLS, add "phylogenetically informed."

Box 2 In general, REML is better than ML in these situations. Needs discussion.

Tables I could not find any. We need all of the stats comparisons in a table.

Figure 1 It's really hard to glean anything from this. Lines are too small, colors are too faint, font is too small, etc. Please find a way to make this legible and convey messages simply.

Figure 6 and others Wow, these colors are just too washed out. We need bolder colors, and good old black, open circle, and gray when possible. Also, squares and triangles as needed.

(Remarks on code availability)

Reviewer #2

(Remarks to the Author)

The authors "adapted" a PGLS model to account for phylogenetic autocorrelation in the dependent variable rather than in the model residuals, referring to predictions derived from this "adapted PGLS" as phylogenetically-informed predictions. They then computed traditional PGLS, which accounts for phylogenetic autocorrelation in the error term, as well as OLS models. The three models (adapted-PGLS, traditional-PGLS, and OLS) were compared in terms of their performance in predicting unknown continuous traits across varying scenarios of tree balance and stemminess. While the method applied in the manuscript was previously proposed by Garland and Ives (2000), the novelty of study lies in exploring how adapted-PGLS performs under different scenarios.

While I commend the authors for a well-written manuscript and their efforts to conduct computationally intensive simulations, an important factor is missing from their sensitivity analyses: tree size (sampling units). Although they tested thousands of simulated phylogenies, they limited tree size to 100 species across all simulations. This value seems relatively low for studies relying on phylogenetic comparative methods (PCM), as PCM research often involves hundreds or even thousands of species. Simulations incorporating different tree sizes would be highly beneficial, as variance in predictions may increase with sample size (along with branch length).

In their simulations, the authors randomly assign missing values to the species traits, but this scenario is rarely encountered

in real data. The literature on missing data mechanisms (MCAR, MAR, MNAR; see <https://doi.org/10.2307/2335739>, <https://doi.org/10.1007/s00265-010-1044-7>) suggests that missing data is most often MNAR. In such cases, phylogenetic autocorrelation may play a more significant role if missingness is related to shared ancestry. It would be interesting to explore a scenario where missing data is restricted to a specific range (e.g., the lower quartile) of simulated continuous traits and to assess how method choice performs under this condition.

Additionally, the authors could enhance the manuscript by including supplementary plots that demonstrate the importance of "method choice" in explaining variation across simulations. For example, Venn diagrams could illustrate the proportion of variation attributable to "methods," "tree balance," "speciation rate," and "correlated evolution rate."

Minor Comments:

Lines 98–99: Are the median prediction errors statistically equivalent across the investigated methods? Please clarify.

Line 105: Clarify what you mean by "perform about 3.5–4x better." I assume this refers to the range of your error metric rather than its mean or median (which appears close to zero across all methods). Similar clarification is needed for lines 108–109, 118–119, 127–128, etc.

Lines 140–144: What is the contribution of tree balance to explaining variation in the median absolute error of predictions from traditional-PGLS and OLS models?

Line 192: Explicitly define "model performance" by describing the evaluation metric used.

Line 702: Does "lambda" refer to Pagel's lambda? Please clarify.

Lines 787–789: Verify whether the referenced study (ref 47) mentions the use of 18 or 19 species absent from the tree, as conflicting values are reported.

Figure 3: Include a test comparing the medians across the three categories in the boxplots. Also, specify in the caption that the squared-sigma symbol represents variance, not the median of the y-axis metric. Apply the same clarification to Extended Data Fig. 1.

Figure 4: The term "grey points" might lead readers to interpret the description as literal points rather than as a scatterplot. Rephrase for clarity.

Figure 5: To facilitate comparison of methods, arrange the histograms vertically in a single column, sharing the x-axis.

Extended Data Fig. 2: Use a consistent color scheme across figures (e.g., always depict the OLS method in green and the phylogenetically-informed method in blue).

(Remarks on code availability)

Reviewer #3

(Remarks to the Author)

This paper explores the topic of prediction in evolutionary biology and makes a strong case that a long-standing, but underutilized approach (that of phylogenetically-informed prediction: Garland and Ives 2000) should be more widely used. Through an extensive series of simulated and empirical examples the authors clearly demonstrate that phylogenetically-informed predictions outperform alternative linear model prediction approaches used with comparative data. They also explain why more standard predictions from common PGLS implementations are better than ignoring the phylogeny, but are still outperformed by phylogenetically-informed GLS prediction. Essentially, the latter makes better use of the position of the taxon on the phylogeny during prediction, thereby improving evolutionary predictions over PGLS (which makes predictions based on the phylogenetic average). This is an important point for empiricists to heed.

This is an important paper and one that provides a strong re-emphasis of the fact that predictions with comparative data should be made in a phylogenetic context, using phylogenetically-informed prediction procedures (sensu Garland and Ives 2000). It is a message that biologists of all stripes should follow.

(Remarks on code availability)

It is sufficient for users to follow

Version 1:

Reviewer comments:

Reviewer #1

(Remarks to the Author)

I have read over the responses of the authors to all three reviewers (one of which was only brief comments). They have been

very responsive and have done an excellent job in revising. All of the issues raised have now been clarified, and substantial materials have been added to the manuscript. The figures look good and the new boxes are very helpful. I have no further comments.

(Remarks on code availability)

Reviewer #2

(Remarks to the Author)

Dear authors,

I appreciate your thoughtful revisions and commend the effort to address the critiques raised in my review. You have now incorporated analyses with varying tree sizes and more realistic missing-data scenarios (MNAR), significantly strengthening the manuscript's robustness. The additional sensitivity analyses and supplementary plots further clarify the role of methodological choices across simulation conditions.

I have no major comments on the current version and believe the manuscript is much improved. Congratulations on this rigorous update. I let below a couple of minor points to consider ahead:

Replace "x" by the proper 'times' symbol.

Figure 5. Please, add a legend informing the color-ramp meaning.

Line 410: Pagel's Lambda? If so, add "Pagel's ".

Figure 6. Have you tried to artificially delete call frequency for some species and then predict it back on these missing data entries and compare such predictions against the observed (omitted) values? You are comparing predictions across approaches, but what about predictions vs observed data?

(Remarks on code availability)

REVIEWER COMMENTS

Reviewer #1 (Remarks to the Author):

Reviewer 1, Comment 1: Good paper in general, but hard to follow. I'll resist saying "hard to see the forest for the trees" in some places. Needs some summary statistical tables.

This is a very helpful suggestion. We have now added tables with summary statistics for each of our simulation analyses (please see Supplementary Tables S1–S13). We have added them to our supplementary information given the large number of tables, but we refer to them throughout the text.

Reviewer 1, Comment 2: Title and elsewhere: I don't think Phylogenetically-Informed needs a hyphen because the adjective ends in lly.

Abstract: "simulations" should be used here so the readers has some idea of what was done

We have now removed the hyphen from the phrase in the title, the abstract, and throughout the manuscript. We have also incorporated our simulations into the new version of the abstract. See also our reply to Reviewer 1, Comment 5.

Reviewer 1, Comment 3: Line 21 simply? Hahaha

We have now removed the word 'simply' from this sentence in our revised abstract.

Reviewer 1, Comment 4: 25 I am confused because PGLS can be used to make "phylogenetically informed" predictions. What's the distinction?

We have now clarified the distinction in the revised abstract, introduction, and Box 1. Please also see our response to Reviewer 1, Comment 11.

Reviewer 1, Comment 5: 27 finish the thought ... by use of what?

Our newly revised abstract now explicitly explains that we use a comprehensive simulation approach in combination with case studies using real data. Please see also our reply to Reviewer 1, Comment 2.

Reviewer 1, Comment 6: 28 correlated with what? do you mean two traits being correlated or do you mean that a given trait (or the residuals) possess phylogenetic signal (sensu Blomberg et al. 2003)?

Here, we refer to the specific correlation between two traits. This is now clarified in our revised abstract (see line 33 in Word document with tracked changes hidden).

Reviewer 1, Comment 7: 32-33 absolutely 35 spelling?

'Palaeontology' is the British English spelling.

Reviewer 1, Comment 8: 55 I think you mean predict the value of a single trait, e.g., body size, as opposed to, say, predicting a Y (e.g., brain size) from an X variable (e.g., body size). Please clarify.

Yes, the reviewer is correct. Phylogenetically informed predictions can be made for a single trait from information about shared ancestry alone or in combination with trait data for known individuals/species. We now clarify this in the revised manuscript. Please see lines 66–68 in Word document with tracked changes hidden.

Reviewer 1, Comment 9: In general, this prose might be made more tangible and relatable with some examples sprinkled in.

This is an excellent suggestion. We have revised our introduction to include explicit and tangible examples. We did this in combination with the revisions outlined above for Reviewer 1, Comment 8. Please see the revised paragraph (lines 49–75).

Reviewer 1, Comment 10: 64 Good point, and others could be cited from the allometry world.

Thank you for this comment. In line with this suggestion, we have now re-worded this section slightly to emphasize the particular abundance of such approaches in the allometry world. We have also added additional references related to allometric predictive equations. Please see lines 79–82.

Reviewer 1, Comment 11: 65-66 confusing terminology/prose. We are talking about predictive equations (or procedures) in both cases.

69-70 Again, confusing, you are slamming PGLS predictive procedures or exalting them?

We apologize for the confusion. To clarify, predictive equations can be generated using either non-phylogenetic regression (OLS) or phylogenetic (PGLS) regression models. In both cases, we can use the predictive equation ($y = \beta x + \alpha$) to “calculate” an unknown value y given the value of the independent variable x . Whilst phylogenetically informed prediction also uses a PGLS regression model, what is notably different is the incorporation of the phylogenetic position of the *unknown* species into the prediction of y – the inference is no longer made by plugging a value into an equation, but instead takes into consideration the position within the variance-covariance matrix of the species used to inform the predictive relationship. We have now made substantial revisions to our text to make this distinction clearer, particularly in lines 76–93 and in Box 1. In combination with our other revisions, we hope that this is now more clearly outlined in the new version of the manuscript.

Reviewer 1, Comment 12: 83 "extant trees" sound strange. Maybe "trees with only extant taxa"

We have now changed this wording in the revised manuscript and explicitly defined “ultrametric” at its first mention (lines 96–97). We similarly define non-ultrametric in

the same part of the text and now use this terminology to refer to trees throughout the manuscript to improve consistency and clarity.

Reviewer 1, Comment 13: 90 Only used ultrametric? I was expecting some paleo relevance, which would use trees with some branches terminating before the present.

Oh, OK, I see it's down below.

As noted by the reviewer, we describe our results on ultrametric and non-ultrametric trees in separate sections. However, to avoid confusion, we now mention both sets of analyses in the introduction, as well as highlighting the incorporation of fossils into our case studies (see lines 94–104).

Reviewer 1, Comment 14: 94 Multivariate? Technically it is also bivariate, so I'd just say that.

We have modified the wording in the revised manuscript in accordance with the reviewer's suggestion (see line 117).

Reviewer 1, Comment 15: 112 Where are the stats shown? Refer to a table?

We have now added new supplementary tables summarising the statistics from the balance and stemminess analyses (see Supplementary Table S3).

Reviewer 1, Comment 16: 137-140 This is a cool result.

We thank the reviewer for noting this – we think so too!

Reviewer 1, Comment 17: 187 Same as setting them all to unity, right? Might say this as well. This is what you say on 285.

Yes, this is true. All branches were set to equal a length of 1. We have now clarified this in the revised manuscript (see lines 283-284).

Reviewer 1, Comment 18: 188 Well, setting branch lengths to anything IS an assumption for the subsequent analyses!!!

We agree with the reviewer that setting all branches equal to 1 is an assumption and clarify it as such in the revised manuscript (see lines 284-286).

Reviewer 1, Comment 19: 213 Avoid starting sentences with "There have been" because (1) it makes "there" the subject of the sentence and (2) is unnecessarily wordy. Try something like "Inferences about how neonatal primate brain size has evolved ..."

Please check for other such instances.

We have used the specific phrasing suggested by the reviewer for this example (line 328). We have now amended similarly structured sentences throughout the revised manuscript.

Reviewer 1, Comment 20: 285 Some other approaches for coming up with arbitrary "starter" branch lengths are also common and should at least be mentioned:

Grafen, A. 1989. The phylogenetic regression. *Philosophical Transactions of the Royal Society of London. Series B, Biological Sciences* 326:119–157.

Garland, Jr., T., P. H. Harvey, and A. R. Ives. 1992. Procedures for the analysis of comparative data using phylogenetically independent contrasts. *Systematic Biology* 41:18–32.

Reviewer 1, Response 20: We have added these references and now expand the text to mention alternative approaches to assign arbitrary branch lengths. Please see lines 400-402.

Reviewer 1, Comment 21: 289 These lambda, alpha and other symbols that are Greek to me come out of nowhere with no explanation. Please explain. Are we talking about Pagel's lambda? What about OU or ACDC transforms, as in Blomberg et al. 2003?

We provide brief explanations of the statistical notation in the revised manuscript. Alpha and beta refer to the parameters of the regression model and are now explicitly defined at first mention (see lines 332-333). Pagel's lambda was indeed used to estimate phylogenetic signal – and this is now described and cited in the methods (lines 585-587) and its first mention in the results (line 336). Phylogenetically informed prediction can be used alongside any homogenous tree transform such as those mentioned by the reviewer. For example, as described in the Methods, we set all branches equal to length 1 using a kappa transform ($\kappa = 0$). We have now incorporated this information into Box 2.

Reviewer 1, Comment 22: 289 Also, I may have missed it, but what software was used to do all of this?

We conducted all phylogenetically informed predictions and simulations using the R scripts provided as Supplementary Data 1. All other analyses were performed in R, using the packages cited in the Methods and detailed in the supplementary files. The description of these methods was originally described in full for the simulations, but we have now added some extra detail to the Methods section for the case studies for clarity. We have also incorporated this information into our Data and Code Availability section.

Reviewer 1, Comment 23: 397 comparative biological studies

We have made this correction in the revised manuscript.

Reviewer 1, Comment 24: 398-401 OK, now I think I get the distinction you are making between PGLS (which I have always considered "phylogenetically informed" because they are!) and what you are calling "phylogenetically informed" here. This is not great terminology, and whatever your distinction should be made clear in the Abstract and Introduction.

We thank the reviewer for helping us make our manuscript clearer. We have made many changes throughout the manuscript in order to explicitly outline our terminology and the distinctions between the approaches. To clarify, whilst the PGLS model is indeed phylogenetically informed, predictions made from the regression model parameters are not. We now outline this in more detail in our abstract, introduction, and Box 1. Please see also our response to Reviewer 1, Comment 11 in which we provide a more detailed description.

Reviewer 1, Comment 25: Box 1 Need to use "PGLS" here

Need to label the three clearly as OLS, PGLS, add "phylogenetically informed."

We have now updated the wording used in the box to better reflect the terminology used throughout the text. We have also updated this figure, including the labelling of the three methods to facilitate quick reference and comparison.

Reviewer 1, Comment 26: Box 2 In general, REML is better than ML in these situations. Needs discussion.

We have now expanded our section where we mention alternative approaches to predictive modelling to include REML and Bayesian models. However, we do not feel that a discussion of REML vs. ML approaches is necessary given the purpose of the box, which is to illustrate that there is no "one size fits all" approach.

Reviewer 1, Comment 27: Tables I could not find any. We need all of the stats comparisons in a table.

We have now incorporated summary tables of the statistics as suggested (see Supplementary Tables S1-S13). This includes additional tables associated with our new simulation results incorporated in response to Reviewer 2, Comments 2-3.

Reviewer 1, Comment 28: Figure 1 It's really hard to glean anything from this. Lines are too small, colors are too faint, font is too small, etc. Please find a way to make this legible and convey messages simply.

Figure 6 and others Wow, these colors are just too washed out. We need bolder colors, and good old black, open circle, and gray when possible. Also, squares and triangles as needed.

We now provide updated versions of our figures incorporating all suggestions made by the reviewer. We have also modified our other figures to retain consistency across the manuscript (see also our response to Reviewer 2, Comments 5 and 14).

Reviewer #2 (Remarks to the Author):

Reviewer 2, Comment 1: The authors "adapted" a PGLS model to account for phylogenetic autocorrelation in the dependent variable rather than in the model residuals, referring to predictions derived from this "adapted PGLS" as phylogenetically-informed predictions. They then computed traditional PGLS, which accounts for phylogenetic autocorrelation in the error term, as well as OLS models. The three models (adapted-PGLS, traditional-PGLS, and OLS) were compared in terms of their performance in predicting unknown continuous traits across varying scenarios of tree balance and stemminess. While the method applied in the manuscript was previously proposed by Garland and Ives (2000), the novelty of study lies in exploring how adapted-PGLS performs under different scenarios.

We thank the reviewer for their concise summary of our analyses, as well as their comments that helped improve this manuscript.

Reviewer 2, Comment 2: While I commend the authors for a well-written manuscript and their efforts to conduct computationally intensive simulations, an important factor is missing from their sensitivity analyses: tree size (sampling units). Although they tested thousands of simulated phylogenies, they limited tree size to 100 species across all simulations. This value seems relatively low for studies relying on phylogenetic comparative methods (PCM), as PCM research often involves hundreds or even thousands of species. Simulations incorporating different tree sizes would be highly beneficial, as variance in predictions may increase with sample size (along with branch length).

Reviewer 2, Response 2: This is a great suggestion. We repeated our simulations for each tree type with $n = 50, 250, \text{ and } 500$ taxa. Overall, we found that the variance in prediction errors decrease for all three methods with larger sample sizes; however, phylogenetically informed predictions still perform better than OLS and PGLS predictive equations (*performance* defined as the fold reduction in the variance of prediction errors) and are more accurate (i.e., have smaller absolute prediction errors) on average. These new results are summarised in our Simulation section, with more detailed results provided in Supplementary Tables S4-S5.

Reviewer 2, Comment 3: In their simulations, the authors randomly assign missing values to the species traits, but this scenario is rarely encountered in real data. The literature on missing data mechanisms (MCAR, MAR, MNAR; see <https://doi.org/10.2307/2335739>, <https://doi.org/10.1007/s00265-010-1044-7>) suggests that missing data is most often MNAR. In such cases, phylogenetic autocorrelation may play a more significant role if missingness is related to shared ancestry. It would be interesting to explore a scenario where missing data is restricted to a specific range (e.g., the lower quartile) of simulated continuous traits and to assess how method choice performs under this condition.

This is a fair comment. We have therefore carried out an additional set of simulations in line with the reviewer's suggestion. In these new simulations, we test an extreme scenario in which all 'unknown values' are explicitly restricted to fall only within the lower quartile of the simulated trait data. We do this for both ultrametric and non-ultrametric trees and report the results in a new figure (Figure 3) and Supplementary Tables S12-S13. We find that prediction errors are negatively biased for all three approaches (larger predictions than actual); however, the phylogenetically informed predictions are less biased and more accurate on average. This result is now

mentioned in our results and discussion (see lines 299-315 in Word document with tracked changes hidden).

Reviewer 2, Comment 4: Additionally, the authors could enhance the manuscript by including supplementary plots that demonstrate the importance of "method choice" in explaining variation across simulations. For example, Venn diagrams could illustrate the proportion of variation attributable to "methods," "tree balance," "speciation rate," and "correlated evolution rate."

In our revised manuscript it is possible to examine the effects of tree balance, correlation value, and prediction method both in our figures (Figures 1-3) and new tables (Supplementary Tables S1-S13). Each table addresses a different aspect of what might impact prediction accuracy or performance. For example, we present the tree balance results in Supplementary Table S3 and the tree size results in Supplementary Tables S4-S5. With regards to method choice, we do not recommend the use of any other approach than phylogenetically informed prediction. Given this, we do not think it is appropriate to compare the simulated variance across different methods in this way. Regardless of tree size or shape, phylogenetically informed predictions are more accurate than predictive equations from either OLS or PGLS.

Minor Comments:

Reviewer 2, Comment 5: Lines 98–99: Are the median prediction errors statistically equivalent across the investigated methods? Please clarify.

We thank the reviewer for seeking clarification on this. The median prediction errors show the level of bias in the simulated datasets. All three methods centre around a median prediction error of 0 because the target taxa were selected at random. Please refer to our new analyses (Figure 3) in response to Reviewer 2, Comment 3, where we simulate a negative bias in the prediction errors.

The distributions of prediction errors shown in our figures (Figures 1 and 3, Extended Data Figures 1 and 3) encompass multiple predictions from multiple trees. To statistically compare the accuracy of the phylogenetically informed predictions and the predictive equations, we calculated differences in the absolute prediction errors between the methods for each prediction made (*error difference* = OLS or PGLS absolute prediction error – phylogenetically informed absolute prediction error). If the difference in error between two methods is positive, then the phylogenetically informed predictions are more accurate (i.e., phylogenetically informed predictions deviate less from the actual values on average). We took the median error difference for each tree and tested if these median error differences were statistically greater than 0 on average across the 1000 trees using an intercept-only linear model. For every tree type and dataset, we found that these median error differences were significantly greater than 0, demonstrating that the phylogenetically informed predictions are more accurate on average.

Reviewer 2, Comment 6: Line 105: Clarify what you mean by "perform about 3.5–4x better." I assume this refers to the range of your error metric rather than its mean or median (which

appears close to zero across all methods). Similar clarification is needed for lines 108–109, 118–119, 127–128, etc.

We thank the reviewer for seeking clarification on this. Here, we use the phrase “performs better” to refer to greater overall performance, as defined in lines 124-127 of our revised manuscript (Word document with tracked changes hidden). In short, our measure of overall performance for a particular method is based on the variance in the distribution of prediction errors. If a method produces a set of predictions with errors that are more widely distributed (or further away from the known value), then this method has a lower overall performance (or “performs worse”). The specific values, such as “a method performs 3.5x better”, are calculated from a comparison of two variances. For example, in Figure 1b, the phylogenetically informed prediction yielded a distribution of prediction errors with a variance of 0.007. The OLS-derived equations yielded a variance of 0.03, which is about 4x greater than 0.007 ($0.03/0.007 = 4.3$). The phylogenetically informed prediction method produced a set of prediction errors that were 4x closer to the actual values overall (i.e., “better”) than the OLS-derived equations. We have now outlined this definition in more detail and incorporated the above example into our revised manuscript (see also lines 124-132) in order to clarify.

Reviewer 2, Comment 7: Lines 140–144: What is the contribution of tree balance to explaining variation in the median absolute error of predictions from traditional-PGLS and OLS models?

We have now added results on the effect of tree balance on absolute prediction error for the PGLS-derived equations (see lines 166-167 and 206-211, and Supplementary Table S3). However, tree balance is not pertinent to OLS prediction errors, as phylogenetic information was not used to make the predictions or the predictive equations.

Reviewer 2, Comment 8: Line 192: Explicitly define “model performance” by describing the evaluation metric used.

We have now fully and explicitly outlined what is meant by our predictive performance metric (see lines 124-127). In our study, performance is measured by comparing the variances in prediction errors. Please see also our reply to Reviewer 2, Comment 6.

Reviewer 2, Comment 9: Line 702: Does “lambda” refer to Pagel's lambda? Please clarify.

The reviewer is correct, and we have updated our manuscript to clarify. Please also see our response to Reviewer 1, Comment 21: “Pagel’s lambda was indeed used to estimate phylogenetic signal – and this is now described and cited in the methods (lines 585-587) and its first mention in the results (line 336).”

Reviewer 2, Comment 10: Lines 787–789: Verify whether the referenced study (ref 47) mentions the use of 18 or 19 species absent from the tree, as conflicting values are reported.

We thank the reviewer for pointing out the discrepancy. The number of species absent from the bush-cricket tree was 18. We have corrected this in our Methods. In the original study, there are multiple taxa that are phylogenetic extrapolations (i.e., fall outside the bush-cricket clade). We only included one of these species in our analyses (see new Figure 6). This may explain differences in taxa numbers reported in our paper and in the original referenced study.

Reviewer 2, Comment 11: Figure 3: Include a test comparing the medians across the three categories in the boxplots. Also, specify in the caption that the squared-sigma symbol represents variance, not the median of the y-axis metric. Apply the same clarification to Extended Data Fig. 1.

We have now included statistics comparing the prediction errors across the three methods – please see our response to Reviewer 2, Comment 5. We have also now specified in the caption that the squared-sigma symbol represents the variance in prediction errors. We have updated both figures (now Figures 1 and 4) in accordance with these suggestions. The specific values are now moved to the Supplementary Tables (for the simulations) and Figure 4 caption.

Reviewer 2, Comment 12: Figure 4: The term "grey points" might lead readers to interpret the description as literal points rather than as a scatterplot. Rephrase for clarity.

Thank you. This sentence was removed to avoid confusion.

Reviewer 2, Comment 13: Figure 5: To facilitate comparison of methods, arrange the histograms vertically in a single column, sharing the x-axis.

We have modified this figure (now Figure 6) in line with this suggestion. Please see the new version of the figure.

Reviewer 2, Comment 14: Extended Data Fig. 2: Use a consistent color scheme across figures (e.g., always depict the OLS method in green and the phylogenetically-informed method in blue).

We have now updated the figure to use a consistent colour scheme as the rest of the figures.

Reviewer #3 (Remarks to the Author):

Reviewer 3, Comment 1: This paper explores the topic of prediction in evolutionary biology and makes a strong case that a long-standing, but underutilized approach (that of phylogenetically-informed prediction: Garland and Ives 2000) should be more widely used. Through an extensive series of simulated and empirical examples the authors clearly demonstrate that phylogenetically-informed predictions outperform alternative linear model prediction approaches used with comparative data. They also explain why more standard predictions from common PGLS implementations are better than ignoring the phylogeny, but are still outperformed by phylogenetically-informed GLS prediction. Essentially, the latter

makes better use of the position of the taxon on the phylogeny during prediction, thereby improving evolutionary predictions over PGLS (which makes predictions based on the phylogenetic average). This is an important point for empiricists to heed.

This is an important paper and one that provides a strong re-emphasis of the fact that predictions with comparative data should be made in a phylogenetic context, using phylogenetically-informed prediction procedures (sensu Garland and Ives 2000). It is a message that biologists of all stripes should follow.

Reviewer #3 (Remarks on code availability):

It is sufficient for users to follow

We appreciate the reviewer for their positive remarks on our paper. We hope others will agree!

REVIEWER COMMENTS

Reviewer #1 (Remarks to the Author):

I have read over the responses of the authors to all three reviewers (one of which was only brief comments). They have been very responsive and have done an excellent job in revising. All of the issues raised have now been clarified, and substantial materials have been added to the manuscript. The figures look good and the new boxes are very helpful. I have no further comments.

We thank the reviewer for their help improving this manuscript.

Reviewer #2 (Remarks to the Author):

Reviewer 2, Comment 1: I appreciate your thoughtful revisions and commend the effort to address the critiques raised in my review. You have now incorporated analyses with varying tree sizes and more realistic missing-data scenarios (MNAR), significantly strengthening the manuscript's robustness. The additional sensitivity analyses and supplementary plots further clarify the role of methodological choices across simulation conditions.

I have no major comments on the current version and believe the manuscript is much improved. Congratulations on this rigorous update. I let below a couple of minor points to consider ahead:

We thank the reviewer for their help improving this manuscript.

Reviewer 2, Comment 2: Replace "x" by the proper 'times' symbol.

We have replaced “x” with the multiplication symbol.

Reviewer 2, Comment 3: Figure 5. Please, add a legend informing the color-ramp meaning.

We have now added a colour scale to Figure 5.

Reviewer 2, Comment 4: Line 410: Pagel's Lambda? If so, add "Pagel's ".

We have added “Pagel’s” here to clarify.

Reviewer 2, Comment 5: Figure 6. Have you tried to artificially delete call frequency for some species and then predict it back on these missing data entries and compare such predictions against the observed (omitted) values? You are comparing predictions across approaches, but what about predictions vs observed data?

We appreciate the reviewer’s suggestion; however, this was the purpose of the first case study. In our first case study, we apply a self-validation approach to assess the accuracy of predicting neonatal brain sizes for taxa that have real observations (n = 45). We then compare the prediction accuracy across the three methods. This self-validation approach demonstrates that phylogenetically informed predictions are more accurate on average than predictive equations (Figure 4c).